# Effect of Reaction Wheel Imbalances on Attitude and Stabilization Accuracy

**Stepan Tkachev** *[ID], **Yaroslav Mashtakov** [ID], **Danil Ivanov** [ID], **Dmitry Roldugin** [ID] and **Mikhail Ovchinnikov** [ID]

Keldysh Institute of Applied Mathematics of RAS, Miusskaya Sq. 4, 125047 Moscow, Russia; yarmashtakov@gmail.com (Y.M.); danilivanovs@gmail.com (D.I.); rolduginds@gmail.com (D.R.); ovchinni@keldysh.ru (M.O.)
* Correspondence: stevens_L@mail.ru

**Abstract:** In this paper, the study of stabilization accuracy of a satellite equipped with a set of reaction wheels (RW) is presented. The model of motion takes into account possible static and dynamic reaction wheel imbalances. Due to the complexity of the model, which leads to the numerical issues, the effects of dynamic and static imbalances on inertial stabilization are studied analytically. As a result, estimations of the attitude and stabilization accuracy are presented in closed form.

**Keywords:** reaction wheels; imbalance; attitude accuracy





## 1. Introduction

Modern small satellites are able to solve many scientific and applied problems, from conducting measurements of the Earth's magnetosphere to remote sensing. Most of these problems require precise satellite pointing and stabilization, which is usually provided by gyroscopic attitude control systems such as Reaction Wheels and Control Moment Gyros. These actuators, thanks to miniaturization, now could be installed even on-board the nanosatellites like 3–6U CubeSats [1–4].

RWs offer rather good performance and accuracy. However, there are several aspects that should be taken into account. The first is the saturation problem, which can be solved by auxiliary actuators such as magnetorquers. The second problem is the vibrations affecting spacecraft hardware, its flexible parts and overall dynamics [5,6], which might be crucial for some missions. One of the vibration sources is the RW imbalance. Typical angular rate of RW is several thousand rotations per minute, therefore, even the small imperfections in their balancing might significantly affect the performance of the attitude control system. There are two types of imbalances that are usually distinguished. The first is the static one, which appears when the RW center of mass is not located at its rotation axis. The second is the dynamic imbalance, which corresponds to the misalignment between the RW principal axis of inertia and the rotation axis (see Figure 1).

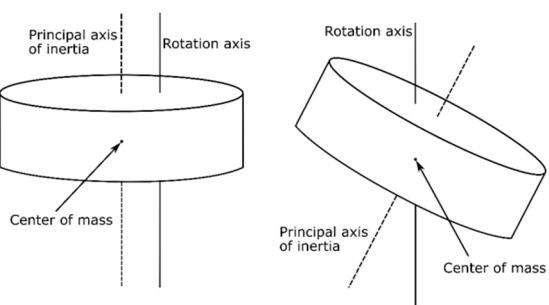

**Figure 1.** RW imbalances: Static (**left**) and dynamic (**right**).

There are a number of approaches that are used to reduce RW vibrations' influence. In the study of [7], small and low-speed RWs are used. In the study of [8], a special platform is proposed and studied. A similar platform and mass dampers are described by the authors of [9–11]. In the study of [12], a two-stage scheme is proposed for the satellite with precise optical payload. In general, different vibration isolation techniques are presented in the survey in [13]. The RW construction improvement is also considered. For example, in the study of [14], the results of liquid RWs in flight operation are presented, and they show low vibration level.

All abovementioned methods require special devices that should be installed on the satellites. Unfortunately, these approaches are usually unavailable for small satellite developers. This problem gets worse due to two additional factors. First, the masses and sizes of the RW and spacecraft are comparable, the reaction wheels might occupy 0.5–1U of 3–6U CubeSat. Second, CubeSat RWs are usually rather cheap, therefore, their requirements might be less strict than for conventional RWs. Therefore, CubeSat developers must pay more attention to the study of stabilization and attitude accuracy.

This paper studies the problem of the attitude and stabilization accuracy in fully coupled motion. The satellite model of motion that takes into account possible RW imbalances is derived. In addition, we investigate how dynamic and static imbalances affect the satellite inertial stabilization, and obtain analytical estimations on possible attitude and stabilization errors. It must be noted that there are several papers that investigate the same problem. For example, in the study of [15], the mathematical model of RW at suspension is presented. The model is rather detailed but the influence of RW imbalances on satellite attitude dynamics is not investigated. Similar results supplemented by experiments are obtained in [16–20]. In the study of [21], the numerical study of the imbalance effect is presented. However, the paper lacks the details. In the study of [22], the model of coupled motion is derived. The detailed numerical study of the satellite angular dynamics is presented and ground tests are carried out. The dynamics model is presented in [23] where the RW installation scheme is proposed to reduce vibrations. However, this model cannot be directly implemented to RW imbalance effect study. The model which is presented in [24] can be adapted to the problem but it requires considerable modification.

The most thorough model of motion is derived in [25] which is based on the first principle and takes into account static and dynamic imbalances. The validation of the model in [25] is carried out using kinetic energy and angular momentum conservation laws. A similar model is derived in the paper, which might be more suitable for the software adaptation. The validation procedure of the derived model is also similar, but the momentum conservation law is added.

The literature study shows the lack of simple expressions for stabilization accuracy. A precise mathematical model is rather complex, and it takes a considerable amount of time to implement it and conduct all the necessary Monte Carlo simulations to estimate the effect of imbalances. This problem occurs because the more angular rate of the RW is—the more effect on the attitude would be. At the same time, high angular rate of the RW requires a rather small integration step. This fact makes it difficult to analyze the dynamics using numerical approach only. There are several ways to deal with this problem. For example, in the study of [26], the general approach to estimate the effect of disturbances on the attitude accuracy is suggested, but it works with a simple model of motion that does not take into account RW imbalances. In the study of [27], the estimations were obtained for the simple case of RW rotation axis deviation. In this paper, the general case of static and dynamic imbalances are under consideration.

The main goal of this paper is to obtain the end form expressions for the attitude and stabilization accuracy, which are useful for preliminary mission design. They require neither program model implementation nor extensive numerical study.

## 2. Preliminary Remarks

Paper [25] describes the coupled model of motion. We present the brief derivation of a similar model here. It has a different final form that turned out to be more convenient for the software implementation and analytical study.

The spacecraft consists of the main hull with several RWs attached. The hull and each RW are supposed to be rigid bodies with known mass and inertia properties. Satellite position is described by point $O$—it is the arbitrary fixed point of the hull. Each reaction wheel is described by point $O_k$ (any point of the RW rotation axis) and axis of rotation $\mathbf{e}_k$ (see Figure 2).

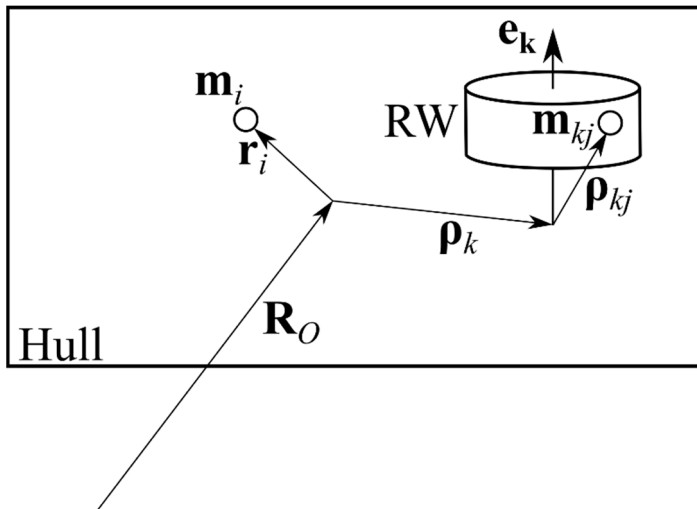

**Figure 2.** Spacecraft and reaction wheel.

The following reference frames are used in the paper. Inertial Frame is fixed in space (e.g., it might correspond to J2000 coordinate system). Body Frame is located at the hull point $O$, and its axes are fixed with respect to the hull, $k$-th RW Frame is attached to the RW in such a way that its third axis corresponds to the rotation axis.

In order to derive the equations of motion the general equation of dynamics [28] for the system with ideal constraints is used:

$$\sum_l \left( m_l \ddot{\mathbf{R}}_l - \mathbf{F}_l \right)^T \delta \mathbf{R}_l = 0 \tag{1}$$

Here, the summation is for all points of the system (for rigid bodies, summation is replaced by integration), $l$ is the system point index, $m_l$ is the point mass, $\ddot{\mathbf{R}}_l$ is its acceleration, $\mathbf{F}_l$ is the total force affecting the point, $\delta \mathbf{R}_l$ is the point virtual displacement.

Every point of the hull $\mathbf{R}_i$ and RW $\mathbf{R}_{kj}$ is described by (see Figure 1):

$$\mathbf{R}_i = \mathbf{R}_O + \mathbf{r}_i,$$
$$\mathbf{R}_{kj} = \mathbf{R}_O + \boldsymbol{\rho}_k + \boldsymbol{\rho}_{kj},$$

where $\mathbf{R}_O$ is the satellite radius-vector, $\mathbf{r}_i$ is the vector from point $O$ to the hull point, $\boldsymbol{\rho}_k = OO_k$, $\boldsymbol{\rho}_{kj}$ is the radius-vector from point $O_k$ to RW point. As was mentioned earlier, $O_k$ is the arbitrary point of RW rotation axis. It is reasonable to choose it as the projection of RW center of mass at its rotation axis. In this case, if the center of mass is on the rotation axis (no static imbalance) then $O_k$ represents the RW center of mass. Index $i$ indicates the

hull point and index $kj$ indicates the $j$-th point of the $k$-th RW in what follows. Virtual displacements of the system points are:

$$\delta\mathbf{R}_i = \delta\mathbf{R}_O + \delta\boldsymbol{\theta} \times \mathbf{r}_i,$$
$$\delta\mathbf{R}_{ki} = \delta\mathbf{R}_O + \delta\boldsymbol{\theta} \times \boldsymbol{\rho}_k + (\mathbf{e}_k\delta\varphi_k + \delta\boldsymbol{\theta}) \times \boldsymbol{\rho}_{kj}.$$

Here, $\delta\mathbf{R}_O, \delta\boldsymbol{\theta}$ correspond to the hull virtual displacements, $\delta\varphi_k$ is the $k$-th RW infinitesimal rotation. Displacements $\delta\mathbf{R}_O, \delta\boldsymbol{\theta}$, $\delta\varphi_k$ are independent and correspond to $N = 6 + n$ system degrees of freedom ($n$ is the RW's quantity, and 6 for translational and rotational degrees of freedom for a rigid hull).

Accelerations of each satellite and RW points are:

$$\ddot{\mathbf{R}}_i = \ddot{\mathbf{R}}_O + \dot{\boldsymbol{\omega}} \times \mathbf{r}_i + \boldsymbol{\omega} \times \boldsymbol{\omega} \times \mathbf{r}_i,$$

$$\ddot{\mathbf{R}}_{kj} = \ddot{\mathbf{R}}_O + \dot{\boldsymbol{\omega}} \times \boldsymbol{\rho}_k + \boldsymbol{\omega} \times \boldsymbol{\omega} \times \boldsymbol{\rho}_k + \left(\dot{\boldsymbol{\omega}} + \dot{\boldsymbol{\Omega}}_k + \boldsymbol{\omega} \times \boldsymbol{\Omega}_k\right) \times \boldsymbol{\rho}_{kj} + (\boldsymbol{\omega} + \boldsymbol{\Omega}_k) \times (\boldsymbol{\omega} + \boldsymbol{\Omega}_k) \times \boldsymbol{\rho}_{kj},$$

where $\boldsymbol{\omega}$ is the satellite angular velocity with respect to the Inertial Frame, $\boldsymbol{\Omega}_k = \dot{\varphi}_k\mathbf{e}_k$ is the RW angular velocity with respect to the hull. Finally, Equation (1) can be rewritten as follows:

$$\sum_i \left(m_i\ddot{\mathbf{R}}_i - \mathbf{F}_i\right)^T\delta\mathbf{R}_i + \sum_k\sum_j \left(m_{kj}\ddot{\mathbf{R}}_{kj} - \mathbf{F}_{kj}\right)^T\delta\mathbf{R}_{kj} = \sum_k M_k^{int}\delta\varphi_k$$

Here, $M_k^{int}$ is the internal torque generated in the $k$-th RW axis that consists of the control and friction torques.

## 3. Equations of Motion Derivation

In order to derive equations of motion, it is necessary to gather all terms for the same virtual displacement and equate them to zero (since they are independent). This leads to the following equations:

$$\delta\mathbf{R}_O : \sum_i\left(m_i\ddot{\mathbf{R}}_i - \mathbf{F}_i\right) + \sum_k\sum_j\left(m_{kj}\ddot{\mathbf{R}}_{kj} - \mathbf{F}_{kj}\right) = 0,$$

$$\delta\boldsymbol{\theta} : \sum_i\mathbf{r}_i \times \left(m_i\ddot{\mathbf{R}}_i - \mathbf{F}_i\right) + \sum_k\sum_j\left(\boldsymbol{\rho}_k + \boldsymbol{\rho}_{kj}\right) \times \left(m_{kj}\ddot{\mathbf{R}}_{kj} - \mathbf{F}_{kj}\right) = 0,$$

$$\delta\varphi_k : \mathbf{e}_k^T\left(\sum_j\boldsymbol{\rho}_{kj} \times \left(m_{kj}\ddot{\mathbf{R}}_{kj} - \mathbf{F}_{kj}\right)\right) - M_k^{int} = 0.$$

These equations can be simplified. The exact derivation is rather bulky and presented in the Appendix A. The final form of the motion equations is provided below.

Let us introduce the cross-product matrix:

$$[\mathbf{a}]_\times := \begin{pmatrix} 0 & -a_3 & a_2 \\ a_3 & 0 & -a_1 \\ -a_2 & a_1 & 0 \end{pmatrix},$$

so $\mathbf{a} \times \mathbf{b} \equiv [\mathbf{a}]_\times\mathbf{b}$. The hull center of mass position with respect to the point $O$ is given by:

$$\mathbf{r}_s = \frac{\sum_i \mathbf{r}_i m_i}{m_s}, \quad m_s = \sum_i m_i.$$

Similarly, the center of mass of $k$-th RW with respect to the point $O_k$ is:

$$\boldsymbol{\rho}_{kc} = \frac{\sum_j \boldsymbol{\rho}_{kj} m_j}{m_k}, \quad m_k = \sum_j m_{kj}.$$

Since the hull point $O$ is arbitrary, it is convenient to choose it, so:

$$m_s \mathbf{r}_s + \sum_k m_k \boldsymbol{\rho}_k = 0 \tag{2}$$

Total satellite mass $m = m_s + \sum_k m_k$. Denote the hull tensor of inertia with respect to the point $O$:

$$\mathbf{J}_s = -\sum_i [\mathbf{r}_i]_\times [\mathbf{r}_i]_\times m_i$$

and $k$-th RW tensor of inertia with respect to the point $O_k$:

$$\mathbf{I}_k = -\sum_j \left[\boldsymbol{\rho}_{kj}\right]_\times \left[\boldsymbol{\rho}_{kj}\right]_\times m_{kj}$$

Total tensor of inertia of the system with respect to the point $O$ is:

$$\mathbf{J} = \mathbf{J}_s + \sum_k \left(-m_k [\boldsymbol{\rho}_k]_\times [\boldsymbol{\rho}_{kc}]_\times - m_k [\boldsymbol{\rho}_{kc}]_\times [\boldsymbol{\rho}_k]_\times - m_k [\boldsymbol{\rho}_k]_\times [\boldsymbol{\rho}_k]_\times + \mathbf{I}_k \right)$$

The total forces acting on the hull and RW are:

$$\mathbf{F}_s = \sum_i \mathbf{F}_i, \quad \mathbf{F}_k = \sum_j \mathbf{F}_{kj}.$$

Torques of the forces affecting RW with respect to the attachment point $O_k$ are:

$$\mathbf{M}_k = \sum_j \boldsymbol{\rho}_{kj} \times \mathbf{F}_{kj}$$

Similarly, the torque affecting the hull with respect to the point $O$ is:

$$\mathbf{M}_s = \sum_i \mathbf{r}_i \times \mathbf{F}_i$$

To decrease the number of brackets in the equations, the following rule is used:

$$\mathbf{a}_1 \times \mathbf{a}_2 \times \ldots \times \mathbf{a}_n \equiv \mathbf{a}_1 \times (\mathbf{a}_2 \times (\ldots \times \mathbf{a}_n)\ldots)$$

Full satellite state vector is described by the hull position $\mathbf{R}_O$, velocity $\mathbf{V}_O = \dot{\mathbf{R}}_O$, attitude quaternion $\mathbf{Q} = \begin{pmatrix} q_0 & \mathbf{q} \end{pmatrix}^T$, $\|\mathbf{Q}\| = q_0^2 + \mathbf{q} \cdot \mathbf{q} = 1$ (optionally can be replaced by different attitude representation, e.g., Euler angles or direction cosine matrices), angular velocity $\boldsymbol{\omega}$, current RW rotation angle $\varphi_k$ and its angular velocity $\Omega_k = \dot{\varphi}_k$. Satellite kinematics are:

$$\begin{aligned} \dot{\mathbf{R}}_O &= \mathbf{V}_O, \\ \dot{q}_0 &= -\tfrac{1}{2}\mathbf{q}^T \boldsymbol{\omega}, \\ \dot{\mathbf{q}} &= \tfrac{1}{2}(q_0 \boldsymbol{\omega} + \mathbf{q} \times \boldsymbol{\omega}), \\ \dot{\varphi}_k &= \Omega_k. \end{aligned} \tag{3}$$

Equations of the satellite dynamics are:

$$\mathbf{S} \begin{pmatrix} \dot{\mathbf{V}}_O \\ \dot{\boldsymbol{\omega}} \\ \dot{\Omega}_1 \\ \vdots \\ \dot{\Omega}_n \end{pmatrix} = \mathbf{N}. \tag{4}$$

Here $\mathbf{S}$ is the following symmetrical matrix:

$$
\mathbf{S} = \begin{pmatrix}
m\mathbf{E}_{3\times3} & -\left[\sum_k m_k\boldsymbol{\rho}_{kc}\right]_\times & -m_1\boldsymbol{\rho}_{1c}\times\mathbf{e}_1 & \cdots & -m_n\boldsymbol{\rho}_{nc}\times\mathbf{e}_n \\
-\left[\sum_k m_k\boldsymbol{\rho}_{kc}\right]_\times & \mathbf{J} & \left(\mathbf{I}_1 - m_1[\boldsymbol{\rho}_1]_\times[\boldsymbol{\rho}_{1c}]_\times\right)\mathbf{e}_1 & \cdots & \left(\mathbf{I}_n - m_n[\boldsymbol{\rho}_n]_\times[\boldsymbol{\rho}_{nc}]_\times\right)\mathbf{e}_n \\
-(m_1\boldsymbol{\rho}_{1c}\times\mathbf{e}_1)^T & \mathbf{e}_1^T\left(\mathbf{I}_1 - m_1[\boldsymbol{\rho}_{1c}]_\times[\boldsymbol{\rho}_1]_\times\right) & I_1 & 0 & 0 \\
\vdots & \vdots & 0 & \ddots & 0 \\
-(m_n\boldsymbol{\rho}_{nc}\times\mathbf{e}_n)^T & \mathbf{e}_n^T\left(\mathbf{I}_n - m_n[\boldsymbol{\rho}_{nc}]_\times[\boldsymbol{\rho}_n]_\times\right) & 0 & 0 & I_n
\end{pmatrix},
$$

where $I_k = \mathbf{e}_k^T\mathbf{I}_k\mathbf{e}_k$ is the RW moment of inertia along its rotation axis, $\mathbf{E}_{3\times3}$ is $3\times3$ identity matrix, and $\mathbf{N}$ is:

$$
\mathbf{N} = \begin{pmatrix}
\mathbf{F}_s + \sum_k \mathbf{F}_k - \sum_k m_k(\boldsymbol{\omega}\times\boldsymbol{\Omega}_k)\times\boldsymbol{\rho}_{kc} - \sum_k m_k(\boldsymbol{\omega}+\boldsymbol{\Omega}_k)\times(\boldsymbol{\omega}+\boldsymbol{\Omega}_k)\times\boldsymbol{\rho}_{kc} \\
\mathbf{N}_\omega \\
M_1^{int} + \mathbf{e}_1^T\mathbf{M}_1 - \mathbf{e}_1^T\left(m_1\boldsymbol{\rho}_{1c}\times\boldsymbol{\omega}\times\boldsymbol{\omega}\times\boldsymbol{\rho}_1 + \mathbf{I}_1(\boldsymbol{\omega}\times\boldsymbol{\Omega}_1) + \boldsymbol{\omega}\times\mathbf{I}_1(\boldsymbol{\omega}+\boldsymbol{\Omega}_1)\right) \\
\vdots \\
M_n^{int} + \mathbf{e}_n^T\mathbf{M}_n - \mathbf{e}_n^T\left(m_n\boldsymbol{\rho}_{nc}\times\boldsymbol{\omega}\times\boldsymbol{\omega}\times\boldsymbol{\rho}_n + \mathbf{I}_n(\boldsymbol{\omega}\times\boldsymbol{\Omega}_n) + \boldsymbol{\omega}\times\mathbf{I}_n(\boldsymbol{\omega}+\boldsymbol{\Omega}_n)\right)
\end{pmatrix},
$$

where

$$
\begin{aligned}
\mathbf{N}_\omega = {} & \mathbf{M}_s + \sum_k(\mathbf{M}_k + \boldsymbol{\rho}_k\times\mathbf{F}_k) - \boldsymbol{\omega}\times\mathbf{J}\boldsymbol{\omega} \\
& -\sum_k\left(\mathbf{I}_k - m_k[\boldsymbol{\rho}_k]_\times[\boldsymbol{\rho}_{kc}]_\times\right)(\boldsymbol{\omega}\times\boldsymbol{\Omega}_k) \\
& -\sum_k m_k(\boldsymbol{\rho}_k\times\boldsymbol{\Omega}_k\times\boldsymbol{\omega}\times\boldsymbol{\rho}_{kc} + \boldsymbol{\rho}_k\times\boldsymbol{\omega}\times\boldsymbol{\Omega}_k\times\boldsymbol{\rho}_{kc} + \boldsymbol{\rho}_k\times\boldsymbol{\Omega}_k\times\boldsymbol{\Omega}_k\times\boldsymbol{\rho}_{kc}) \\
& -\sum_k(\boldsymbol{\Omega}_k\times\mathbf{I}_k\boldsymbol{\omega} + \boldsymbol{\omega}\times\mathbf{I}_k\boldsymbol{\Omega}_k + \boldsymbol{\Omega}_k\times\mathbf{I}_k\boldsymbol{\Omega}_k).
\end{aligned}
$$

Note that the hull is considered to be a rigid body, so $\mathbf{J}_s$ and $\mathbf{r}_s$ remain constant in the Body Frame. However, in general case, $\boldsymbol{\rho}_{kc}$, $\mathbf{I}_k$ and $\mathbf{J}$ depend on the current RW rotation angle $\varphi_k$, and will change in the Body Frame. In addition, satellite acceleration (i.e., $\dot{\mathbf{V}}_O$) is usually calculated in the Inertial Frame, while attitude dynamics are calculated in the Body Frame, which must be taken into account during the simulation.

### 3.1. Important Special Cases

The first special case to be considered is when there is no so-called static imbalance of RW, i.e., its center of mass is located at the rotation axis. In this case, it is reasonable to choose $O_k$ as RW center of mass. Therefore, $O$ under Constraint (2) would denote the center of mass of the whole system, and $\boldsymbol{\rho}_{kc} = 0$. System Kinematics (3) remains the same, and dynamics becomes:

$$
\begin{aligned}
& m\ddot{\mathbf{R}}_O = \mathbf{F}_s + \sum_k \mathbf{F}_k, \\
& \mathbf{J}\dot{\boldsymbol{\omega}} = -\boldsymbol{\omega}\times\mathbf{J}\boldsymbol{\omega} - \sum_k\mathbf{I}_k\dot{\boldsymbol{\Omega}}_k \\
& \qquad -\sum_k(\mathbf{I}_k(\boldsymbol{\omega}\times\boldsymbol{\Omega}_k) + \boldsymbol{\Omega}_k\times\mathbf{I}_k\boldsymbol{\omega} + \boldsymbol{\omega}\times\mathbf{I}_k\boldsymbol{\Omega}_k + \boldsymbol{\Omega}_k\times\mathbf{I}_k\boldsymbol{\Omega}_k) \\
& \qquad +\mathbf{M}_s + \sum_k(\mathbf{M}_k + \boldsymbol{\rho}_k\times\mathbf{F}_k), \\
& \mathbf{e}_k^T\mathbf{I}_k\left(\dot{\boldsymbol{\omega}} + \dot{\boldsymbol{\Omega}}_k\right) = M_k^{int} + \mathbf{e}_k^T\mathbf{M}_k - \mathbf{e}_k^T\left(\mathbf{I}_k(\boldsymbol{\omega}\times\boldsymbol{\Omega}_k) + (\boldsymbol{\omega}+\boldsymbol{\Omega}_k)\times\mathbf{I}_k(\boldsymbol{\omega}+\boldsymbol{\Omega}_k)\right).
\end{aligned}
\tag{5}
$$

If, in addition, there is no dynamical imbalance (i.e., the rotation axis is the RW dynamical symmetry axis), then $\mathbf{I}_k(\boldsymbol{\omega} \times \boldsymbol{\Omega}_k) + \boldsymbol{\Omega}_k \times \mathbf{I}_k\boldsymbol{\omega} = 0$, $\mathbf{I}_k\mathbf{e}_k = I_k\mathbf{e}_k$, $\mathbf{e}_k^T(\boldsymbol{\omega} \times \mathbf{I}_k\boldsymbol{\omega}) = 0$. This leads to the following equations of motion:

$$m\ddot{\mathbf{R}}_O = \mathbf{F}_s + \sum_k \mathbf{F}_k,$$
$$\mathbf{J}\dot{\boldsymbol{\omega}} + \boldsymbol{\omega} \times \mathbf{J}\boldsymbol{\omega} = \mathbf{M}_s + \sum_k (\mathbf{M}_k + \boldsymbol{\rho}_k \times \mathbf{F}_k) - \sum_k \mathbf{I}_k\dot{\boldsymbol{\Omega}}_k - \boldsymbol{\omega} \times \sum_k (\mathbf{I}_k\boldsymbol{\Omega}_k), \tag{6}$$
$$\mathbf{e}_k^T\mathbf{I}_k\left(\dot{\boldsymbol{\omega}} + \dot{\boldsymbol{\Omega}}_k\right) = M_k^{int} + \mathbf{e}_k^T\mathbf{M}_k.$$

In the matrix form:

$$\mathbf{S} = \begin{pmatrix} m\mathbf{E}_{3\times3} & 0 & 0 & \dots & 0 \\ 0 & \mathbf{J} & \mathbf{I}_1\mathbf{e}_1 & \dots & \mathbf{I}_n\mathbf{e}_n \\ 0 & \mathbf{e}_1^T\mathbf{I}_1 & I_1 & 0 & 0 \\ \vdots & \vdots & 0 & \ddots & 0 \\ 0 & \mathbf{e}_n^T\mathbf{I}_n & 0 & 0 & I_n \end{pmatrix}, \quad \mathbf{N} = \begin{pmatrix} \mathbf{F}_s + \sum_k \mathbf{F}_k \\ \mathbf{M}_s + \sum_k \mathbf{M}_{kc} - \boldsymbol{\omega} \times \mathbf{J}\boldsymbol{\omega} - \sum_k \mathbf{I}_k\boldsymbol{\Omega}_k(\boldsymbol{\omega} \times \mathbf{e}_k) \\ M_k + \mathbf{e}_k^T\mathbf{M}_k - \mathbf{e}_k^T(\boldsymbol{\omega} \times \mathbf{I}_k\boldsymbol{\omega}) \end{pmatrix}$$

The simplified conventional model of the satellite motion is (see e.g., [29]):

$$m\ddot{\mathbf{R}}_O = \mathbf{F}_{total},$$
$$\mathbf{J}\dot{\boldsymbol{\omega}} + \boldsymbol{\omega} \times \mathbf{J}\boldsymbol{\omega} = \mathbf{M}_{total} - \sum_k \mathbf{I}_k\dot{\boldsymbol{\Omega}}_k - \boldsymbol{\omega} \times \sum_k \mathbf{I}_k\boldsymbol{\Omega}_k, \tag{7}$$
$$I_k\dot{\boldsymbol{\Omega}}_k = M_k^{int}.$$

It is rather similar to the simplest Model (6), with the major difference in the equation that describes the RW rotation: it is actually affected by satellite rotation and external torques, e.g., by the gravity gradient torque. Note that internal torques from friction and control devices are usually much larger than external ones, so the effect of these additional terms is negligible.

### 3.2. Verification of the Motion Model

The model presented in Section 2 is rather complex. The conservation laws are used to validate its software implementation. In case of no external forces and torques, the total momentum in the Inertial Frame, angular momentum with respect to the system center of mass in Inertial Frame, and kinetic energy of the system must preserve, i.e., they are first integrals.

Total momentum of the system is:

$$\mathbf{P} = \sum_i m_i\mathbf{V}_i + \sum_k\sum_j m_{kj}\mathbf{V}_{kj}$$

Velocities of the satellite points are:

$$\mathbf{V}_i = \mathbf{V}_O + \boldsymbol{\omega} \times \mathbf{r}_i,$$
$$\mathbf{V}_{kj} = \mathbf{V}_O + \boldsymbol{\omega} \times \left(\boldsymbol{\rho}_k + \boldsymbol{\rho}_{kj}\right) + \boldsymbol{\Omega}_k \times \boldsymbol{\rho}_{kj}.$$

Hence, using Constraint (2), we obtain:

$$\mathbf{P} = \left(m_s + \sum_k m_k\right)\mathbf{V}_O + \sum_k m_k(\boldsymbol{\omega} + \boldsymbol{\Omega}_k) \times \boldsymbol{\rho}_{kc}$$

Total angular momentum with respect to the system center of mass is:

$$\mathbf{L}_C = \sum_i (\mathbf{R}_i - \mathbf{R}_C) \times m_i\mathbf{V}_i + \sum_k\sum_j \left(\mathbf{R}_{kj} - \mathbf{R}_C\right) \times m_{kj}\mathbf{V}_{kj}$$

where $\mathbf{R}_C$ is the system center of mass. Under Constraint (2) it can be written as follows:

$$\mathbf{R}_C = \frac{1}{m}\left(m_s(\mathbf{R}_O + \mathbf{r}_s) + \sum_k m_k(\mathbf{R}_O + \boldsymbol{\rho}_k + \boldsymbol{\rho}_{kc})\right) = \mathbf{R}_O + \frac{1}{m}\sum_k m_k \boldsymbol{\rho}_{kc}.$$

In the case when RW center of mass is at the rotation axis $\mathbf{R}_C = \mathbf{R}_O$. Simplification yields:

$$\mathbf{L}_C = \mathbf{J}_s\boldsymbol{\omega} + \sum_k (m_k\boldsymbol{\rho}_k \times (\boldsymbol{\omega} \times (\boldsymbol{\rho}_k + \boldsymbol{\rho}_{kc}) + \boldsymbol{\Omega}_k \times \boldsymbol{\rho}_{kc}) + m_k\boldsymbol{\rho}_{kc} \times \boldsymbol{\omega} \times \boldsymbol{\rho}_k + \mathbf{I}_k(\boldsymbol{\omega} + \boldsymbol{\Omega}_k)) +$$
$$-\frac{1}{m}\left(\sum_k m_k\boldsymbol{\rho}_{kc}\right) \times \left(\sum_k m_k(\boldsymbol{\omega} + \boldsymbol{\Omega}_k) \times \boldsymbol{\rho}_{kc}\right).$$

Total kinetic energy is:

$$T = \frac{1}{2}\sum_i m_i\mathbf{V}_i^2 + \frac{1}{2}\sum_k\sum_j m_{kj}\mathbf{V}_{kj}^2$$

After simplification:

$$T = \frac{1}{2}m\mathbf{V}_O^2 + \frac{1}{2}\boldsymbol{\omega}^T\mathbf{J}_s\boldsymbol{\omega} + \sum_k\left(m_k\mathbf{V}_O^T(\boldsymbol{\omega} + \boldsymbol{\Omega}_k) \times \boldsymbol{\rho}_{kc} + (\boldsymbol{\omega} \times \boldsymbol{\rho}_k)^T((\boldsymbol{\omega} + \boldsymbol{\Omega}_k) \times \boldsymbol{\rho}_{kc})\right) +$$
$$+\frac{1}{2}\sum_k\left((\boldsymbol{\omega} \times \boldsymbol{\rho}_k)^T(\boldsymbol{\omega} \times \boldsymbol{\rho}_k) + (\boldsymbol{\omega} + \boldsymbol{\Omega}_k)^T\mathbf{I}_k(\boldsymbol{\omega} + \boldsymbol{\Omega}_k)\right).$$

The numerical simulation is carried out to demonstrate the behavior of the abovementioned integrals with different integration steps. The system parameters are presented in Table 1. RW parameters are taken from [30].

**Table 1.** System parameters.

| Parameter | Value |
|---|---|
| Hull inertia tensor, $\mathbf{J}_s$ | diag$\begin{pmatrix} 0.027 & 0.03 & 0.01 \end{pmatrix}$ kg·m$^2$ |
| Hull mass, $m_s$ | 3.2 kg |
| RW static imbalance, $s$ | $6.7 \times 10^{-7}$ kg·m |
| RW dynamic imbalance, $d$ | $1.8 \times 10^{-9}$ kg·m$^2$ |
| RW mass, $m_k$ | 0.119 kg |
| RW axis moment of inertia, $I_{ax}$ | $1.67 \times 10^{-4}$ kg·m$^2$ |
| RW equatorial moment of inertia, $I_{eq}$ | $1.0 \times 10^{-4}$ kg·m$^2$ |
| Hull inertia tensor, $\mathbf{J}_s$ | diag$\begin{pmatrix} 0.027 & 0.03 & 0.01 \end{pmatrix}$ kg·m$^2$ |

The set of three identical RWs is considered. Their nominal axes of symmetry are aligned with the hull principal axes of inertia and each center of mass is shifted by 5 cm along x-axis of the Body Frame, i.e., $\boldsymbol{\rho}_k = \begin{pmatrix} 0.05 & 0 & 0 \end{pmatrix}$ m. The initial conditions are presented in Table 2. Here, the satellite free motion is considered, and the initial conditions for translational motion are zero.

**Table 2.** Initial conditions.

| Parameter | Value |
|---|---|
| Satellite angular velocity, $\boldsymbol{\omega}$ | $\begin{pmatrix} 0.14 & 0.34 & 0.23 \end{pmatrix}$ rad/s |
| Quaternion | $\begin{pmatrix} 0.5 & 0.5 & 0.5 & 0.5 \end{pmatrix}$ |
| RWs' angular velocities | $\begin{pmatrix} 10 & 20 & 30 \end{pmatrix}$ rad/s |
| RWs' initial phase | $\begin{pmatrix} 0 & 0 & 0 \end{pmatrix}$ |
| Satellite position | $\begin{pmatrix} 0 & 0 & 0 \end{pmatrix}$ m |
| Satellite velocity | $\begin{pmatrix} 0 & 0 & 0 \end{pmatrix}$ m/s |

Integration method is the fourth-order Runge–Kutta method with constant step. There are two time steps considered: $h = 10^{-2}$ s and $h = 10^{-3}$ s. Results are presented in Figures 3–5, and demonstrate the relative errors of the corresponding values.

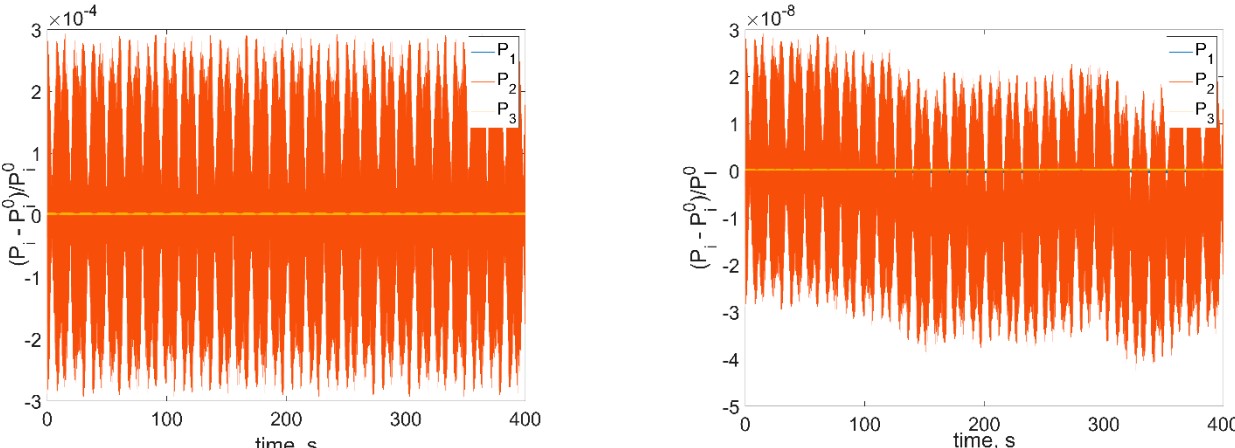

**Figure 3.** Relative variation of the total momentum (**left**) $h = 10^{-2}$ s and (**right**) $h = 10^{-3}$ s).

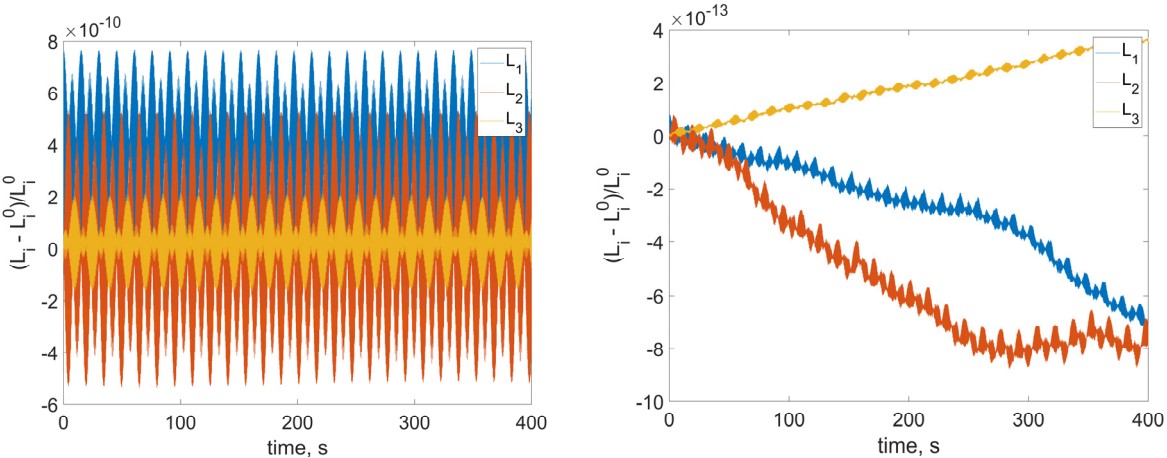

**Figure 4.** Relative variation of the total angular momentum (**left**) $h = 10^{-2}$ s and (**right**) $h = 10^{-3}$ s).

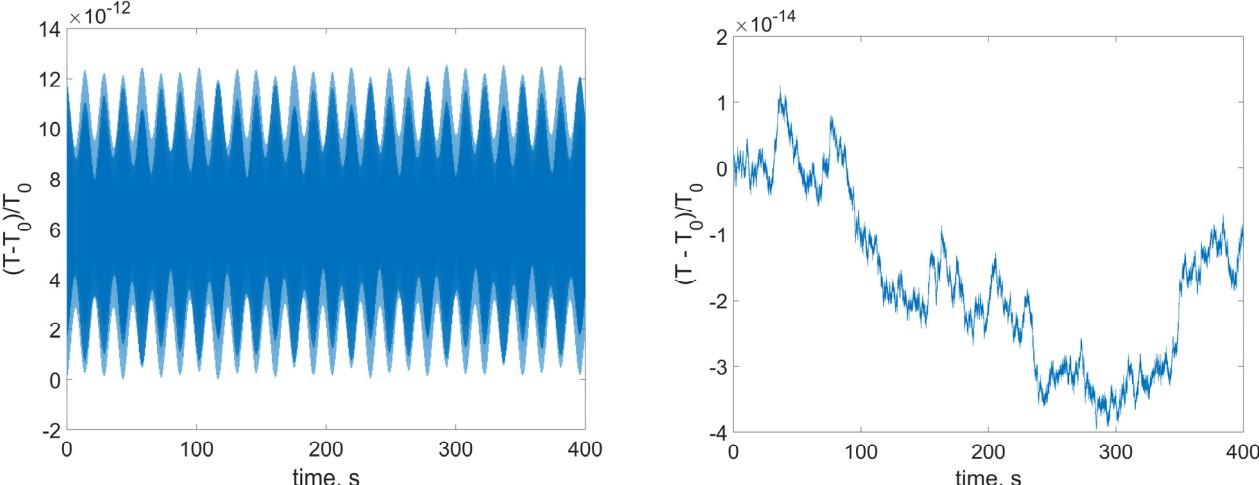

**Figure 5.** Relative variation of the total kinetic energy (**left**) $h = 10^{-2}$ s and (**right**) $h = 10^{-3}$ s).

As we can see from Figures 3–5, the variation of the presented values is lower and more irregular for smaller integration step. So, the numerical model shows the same conservation laws as the mathematical one. Note that in order to correctly simulate the satellite dynamics, it is necessary to consider rather small time step, even for the relatively low RWs angular velocities. In real space applications the RW angular velocity reaches hundreds of radians per second. Additionally, the model includes a number of parameters: inertia tensor, RWs' placement vectors, imbalance and inertia parameters, RWs' angular velocities, as well as control coefficients, which gives at least 24 independent parameters for a satellite with three identical RWs. To have informative enough numerical results, a vast amount of numerical experiments should be carried out. So, the numerical analysis becomes rather computationally expensive. Additionally, the problem of data visualization arises. On the other hand, the analytical estimations on the attitude and stabilization accuracy can be obtained. They allow avoiding extensive numerical simulations during the satellite preliminary design stage. The following sections provide a rather simple end-form expression for the inertial stabilization accuracy.

### 4. Effect of Dynamic Imbalances on Inertial Stabilization

In this section, we analyze the effect of the dynamic imbalances in the case of inertial stabilization when the satellite is in the specific attitude with zero angular velocity. Without loss of generality, let the desired attitude be the identity quaternion $\mathbf{Q}_d = \begin{pmatrix} 1 & 0 & 0 & 0 \end{pmatrix}^T$. All the external torques, as well as RWs' friction, are neglected during the analysis. In addition, we do not consider static imbalance, i.e., RWs' centers of mass are located at their rotation axes.

#### 4.1. First-Order Equations of Motion

The dynamic imbalance appears when RW rotation axis is misaligned with the principal axis of inertia, i.e., there are nondiagonal elements in the RW tensor of inertia $\mathbf{I}_k$. The magnitude of these elements $d$ (such that $\left| I_{kj} \right| < d$ for $k \neq j$) is usually referred to as dynamic imbalance value. It is rather small in comparison with the RW axial moment of inertia $I_{ax}$, therefore, we can introduce small parameter $\varepsilon = d / I_{ax}$. The RW tensor of inertia is then represented in the following way:

$$\mathbf{I}_k = \widetilde{\mathbf{I}}_k + \varepsilon \delta \mathbf{I}_k$$

where $\varepsilon \ll 1$, $\widetilde{\mathbf{I}}_k$ is nominal RW tensor of inertia such that RW rotation axis $\mathbf{e}_k$ is its axis of dynamical symmetry, $\varepsilon \delta \mathbf{I}_k$ is the imbalance additional term. Since $\widetilde{\mathbf{I}}_k$ is axially symmetrical, it does not change in the Body Frame during the RW rotation. Total satellite tensor of inertia in the case of absent static imbalance (i.e., $\boldsymbol{\rho}_{kc} = 0$) is:

$$\mathbf{J} = \mathbf{J}_s + \sum_k \left( \widetilde{\mathbf{I}}_k + \varepsilon \delta \mathbf{I}_k - m_k [\boldsymbol{\rho}_k]_\times [\boldsymbol{\rho}_k]_\times \right) = \widetilde{\mathbf{J}} + \varepsilon \sum_k \delta \mathbf{I}_k$$

Again, here, $\widetilde{\mathbf{J}}$ is the nominal satellite full tensor of inertia, and $\varepsilon \sum_k \delta \mathbf{I}_k$ corresponds to the small deviations caused by imbalances. Usually, these imbalances are unknown, so simplified equations of motions such as Equation (6) with nominal values of RW and satellite tensors of inertia are used to construct attitude controller.

To analyze the effect of the dynamical imbalances consider equations of motion, Equation (5), taking into account the differences between the nominal and real tensors of

inertia. The orbital dynamics are decoupled, in addition, the effect of external torques can be neglected since the time intervals are small. Thus, the system to be considered is:

$$
\mathbf{J}\dot{\boldsymbol{\omega}} + \sum_k \mathbf{I}_k \dot{\boldsymbol{\Omega}}_k = -\boldsymbol{\omega} \times \mathbf{J}\boldsymbol{\omega}
$$
$$
-\sum_k \left( \mathbf{I}_k(\boldsymbol{\omega} \times \boldsymbol{\Omega}_k) + \boldsymbol{\Omega}_k \times \mathbf{I}_k \boldsymbol{\omega} + \boldsymbol{\omega} \times \mathbf{I}_k \boldsymbol{\Omega}_k + \boldsymbol{\Omega}_k \times \mathbf{I}_k \boldsymbol{\Omega}_k \right),
$$
$$
\mathbf{e}_k^T \mathbf{I}_k \left( \dot{\boldsymbol{\omega}} + \dot{\boldsymbol{\Omega}}_k \right) = M_k^{int} - \mathbf{e}_k^T \left( \mathbf{I}_k(\boldsymbol{\omega} \times \boldsymbol{\Omega}_k) + (\boldsymbol{\omega} + \boldsymbol{\Omega}_k) \times \mathbf{I}_k(\boldsymbol{\omega} + \boldsymbol{\Omega}_k) \right),
$$
$$
\dot{\varphi}_k = \Omega_k,
$$
$$
\dot{q}_0 = -\tfrac{1}{2}\mathbf{q}\cdot\boldsymbol{\omega},
$$
$$
\dot{\mathbf{q}} = \tfrac{1}{2}(q_0\boldsymbol{\omega} + \mathbf{q} \times \boldsymbol{\omega}).
$$

To eliminate a small parameter in the left part of the equations, let us rewrite the first two equations as follows:

$$
\left( \widetilde{\mathbf{J}} + \varepsilon \sum_k \delta \mathbf{I}_k \right) \dot{\boldsymbol{\omega}} + \sum_k \left( \widetilde{\mathbf{I}}_k + \varepsilon \delta \mathbf{I}_k \right) \dot{\boldsymbol{\Omega}}_k = \mathbf{a} + \varepsilon \delta \mathbf{a},
$$
$$
\mathbf{e}_k^T \left( \widetilde{\mathbf{I}}_k + \varepsilon \delta \mathbf{I}_k \right) \left( \dot{\boldsymbol{\omega}} + \dot{\boldsymbol{\Omega}}_k \right) = b_k + \varepsilon \delta b_k,
$$

where

$$
\mathbf{a} = -\boldsymbol{\omega} \times \left( \widetilde{\mathbf{J}}\boldsymbol{\omega} + \sum_k \widetilde{\mathbf{I}}_k \boldsymbol{\Omega}_k \right),
$$
$$
\delta \mathbf{a} = -\sum_k (\delta \mathbf{I}_k(\boldsymbol{\omega} \times \boldsymbol{\Omega}_k) + (\boldsymbol{\omega} + \boldsymbol{\Omega}_k) \times \delta \mathbf{I}_k(\boldsymbol{\omega} + \boldsymbol{\Omega}_k)),
$$
$$
b_k = M_k^{int},
$$
$$
\delta b_k = -\mathbf{e}_k^T(\delta \mathbf{I}_k(\boldsymbol{\omega} \times \boldsymbol{\Omega}_k) + (\boldsymbol{\omega} + \boldsymbol{\Omega}_k) \times \delta \mathbf{I}_k(\boldsymbol{\omega} + \boldsymbol{\Omega}_k)).
$$

Finally, this system becomes:

$$
(\mathbf{B} + \varepsilon \delta \mathbf{B}) \begin{pmatrix} \dot{\boldsymbol{\omega}} \\ \dot{\boldsymbol{\Omega}} \end{pmatrix} = \begin{pmatrix} \mathbf{a} \\ \mathbf{b} \end{pmatrix} + \varepsilon \begin{pmatrix} \delta \mathbf{a} \\ \delta \mathbf{b} \end{pmatrix},
$$

$$
\mathbf{B} = \begin{pmatrix} \widetilde{\mathbf{J}} & \widetilde{\mathbf{I}}_1 \mathbf{e}_1 & \cdots & \widetilde{\mathbf{I}}_n \mathbf{e}_n \\ \mathbf{e}_1^T \widetilde{\mathbf{I}}_1 & \mathbf{e}_1^T \widetilde{\mathbf{I}}_1 \mathbf{e}_1 & \cdots & 0 \\ \vdots & \vdots & \ddots & \vdots \\ \mathbf{e}_n^T \widetilde{\mathbf{I}}_n & 0 & \cdots & \mathbf{e}_n^T \widetilde{\mathbf{I}}_n \mathbf{e}_n \end{pmatrix}, \quad \delta \mathbf{B} = \begin{pmatrix} \delta \mathbf{J} & \delta \mathbf{I}_1 \mathbf{e}_1 & \cdots & \delta \mathbf{I}_n \mathbf{e}_n \\ \mathbf{e}_1^T \delta \mathbf{I}_1 & \mathbf{e}_1^T \delta \mathbf{I}_1 \mathbf{e}_1 & \cdots & 0 \\ \vdots & \vdots & \ddots & \vdots \\ \mathbf{e}_n^T \delta \mathbf{I}_n & 0 & \cdots & \mathbf{e}_n^T \delta \mathbf{I}_n \mathbf{e}_n \end{pmatrix}.
$$

The main idea is to represent this system as:

$$
\widetilde{\mathbf{B}} \begin{pmatrix} \dot{\boldsymbol{\omega}} \\ \dot{\boldsymbol{\Omega}} \end{pmatrix} = \begin{pmatrix} \widetilde{\mathbf{a}} \\ \mathbf{b} \end{pmatrix} + \varepsilon \begin{pmatrix} \delta \widetilde{\mathbf{a}} \\ \delta \widetilde{\mathbf{b}} \end{pmatrix} + O\left( \varepsilon^2 \right)
$$

where $\widetilde{\mathbf{B}}$ does not contain a small parameter $\varepsilon$. Detailed derivation is provided in the Appendix B, and here we just present the result:

$$
\left( \widetilde{\mathbf{J}} - \sum_k \widetilde{I}_k \mathbf{e}_k \mathbf{e}_k^T \right) \dot{\boldsymbol{\omega}} = \mathbf{c} + \varepsilon \delta \mathbf{c},
$$
$$
\widetilde{I}_k \dot{\boldsymbol{\Omega}}_k = b_k - \widetilde{I}_k \mathbf{e}_k^T \left( \widetilde{\mathbf{J}} - \sum_k \widetilde{I}_k \mathbf{e}_k \mathbf{e}_k^T \right)^{-1} \mathbf{c}
$$
$$
+ \varepsilon \left( \delta b_k - \frac{\delta I_k}{\widetilde{I}_k} b_k \right)
$$
$$
+ \varepsilon \delta I_k \mathbf{e}_k^T \left( \widetilde{\mathbf{J}} - \sum_k \widetilde{I}_k \mathbf{e}_k \mathbf{e}_k^T \right)^{-1} \mathbf{c} - \varepsilon \mathbf{e}_k^T \delta \mathbf{I}_k \left( \widetilde{\mathbf{J}} - \sum_k \widetilde{I}_k \mathbf{e}_k \mathbf{e}_k^T \right)^{-1} \mathbf{c}
$$
$$
- \varepsilon \widetilde{I}_k \mathbf{e}_k^T \left( \widetilde{\mathbf{J}} - \sum_k \widetilde{I}_k \mathbf{e}_k \mathbf{e}_k^T \right)^{-1} \delta \mathbf{c}.
$$

(8)

where

$$\delta I_k = \mathbf{e}_k^T \delta \mathbf{I}_k \mathbf{e}_k$$
$$\mathbf{c} = \mathbf{a} - \sum_k b_k \mathbf{e}_k,$$
$$\delta \mathbf{c} = \delta \mathbf{a} - \sum_k \left( \delta b_k \mathbf{e}_k + b_k \left( \mathbf{E}_{3\times3} - \mathbf{e}_k \mathbf{e}_k^T \right) \frac{\delta \mathbf{I}_k \mathbf{e}_k}{\widetilde{I}_k} \right)$$
$$- \sum_k \left( \delta \mathbf{I}_k - \delta \mathbf{I}_k \mathbf{e}_k \mathbf{e}_k^T - \mathbf{e}_k \mathbf{e}_k^T \delta \mathbf{I}_k + \mathbf{e}_k \mathbf{e}_k^T \delta I_k \right) \left( \widetilde{\mathbf{J}} - \sum_k \widetilde{I}_k \mathbf{e}_k \mathbf{e}_k^T \right)^{-1} \left( \mathbf{a} - \sum_k b_k \mathbf{e}_k \right).$$

These equations are used later to analyze the satellite motion in the vicinity of the desired position.

*4.2. Controller Design*

Lyapunov-based attitude controller that ensures the convergence to the necessary [31,32] attitude is:

$$\mathbf{M}_{ctrl} = -k_\omega \boldsymbol{\omega} - k_q \mathbf{q} + \boldsymbol{\omega} \times \widetilde{\mathbf{J}} \boldsymbol{\omega}$$

Here, the simplicity of the desired motion (i.e., desired angular velocity and acceleration are equal to zero) is already taken into account. In addition, we consider the case when there are no external disturbances affecting the motion. Note that there is a nominal tensor of inertia in the equation because the attitude control system does not have information about small deviations in the satellite inertia. This is the ideal control torque to be produced by the system of RWs. The necessary RWs' angular accelerations are then defined by:

$$- \sum_k \widetilde{\mathbf{I}}_k \dot{\boldsymbol{\Omega}}_k - \boldsymbol{\omega} \times \left( \sum_k \widetilde{\mathbf{I}}_k \boldsymbol{\Omega}_k \right) = \mathbf{M}_{ctrl} \tag{9}$$

Note that:

$$\widetilde{\mathbf{I}}_k \dot{\boldsymbol{\Omega}}_k = \widetilde{I}_k \dot{\Omega}_k \mathbf{e}_k, \quad \widetilde{\mathbf{I}}_k \boldsymbol{\Omega}_k = \widetilde{I}_k \Omega_k \mathbf{e}_k$$

Introducing:

$$\mathbf{G} = \begin{pmatrix} \widetilde{I}_1 \mathbf{e}_1 & \dots & \widetilde{I}_n \mathbf{e}_n \end{pmatrix}, \quad \boldsymbol{\Omega} = \begin{pmatrix} \Omega_1 & \dots & \Omega_n \end{pmatrix}^T$$

Equation (9) is rewritten as follows:

$$\mathbf{G} \dot{\boldsymbol{\Omega}} = -\mathbf{M}_{ctrl} - \boldsymbol{\omega} \times (\mathbf{G} \boldsymbol{\Omega}) \tag{10}$$

This system of linear equations allows us to calculate the RWs' accelerations using information about current RWs' angular velocities and the necessary control torque. To ensure controllability of the system, it is necessary to install at least three RWs with non-coplanar rotation axes. In this case, Expression (10) has a unique solution:

$$\dot{\boldsymbol{\Omega}} = -\mathbf{G}^{-1} (\mathbf{M}_{ctrl} + \boldsymbol{\omega} \times (\mathbf{G} \boldsymbol{\Omega})) \tag{11}$$

The solution is not unique if there are more than three RWs. In this case, it is reasonable to set an optimization problem:

$$\sum_k \dot{\Omega}_k^2 \to \min$$

under Constraint (10) to reduce total RWs' accelerations. Its solution is well-known and given by the Moore-Penrose pseudoinverse:

$$\dot{\boldsymbol{\Omega}} = -\mathbf{G}^T \left( \mathbf{G} \mathbf{G}^T \right)^{-1} (\mathbf{M}_{ctrl} + \boldsymbol{\omega} \times (\mathbf{G} \boldsymbol{\Omega}))$$

Note that if there are only three RWs, then $\mathbf{G}^T\left(\mathbf{G}\mathbf{G}^T\right)^{-1} = \mathbf{G}^{-1}$, and the solution is the same as in Equation (11). In a simple model RWs' accelerations are:

$$\widetilde{I}_k\dot{\Omega}_k = M_k^{int}$$

Hence, internal control torques are defined by:

$$M_k^{int} = -\widetilde{I}_k^2\mathbf{e}_k^T\left(\mathbf{G}\mathbf{G}^T\right)^{-1}\left(\mathbf{M}_{ctrl} + \boldsymbol{\omega}\times(\mathbf{G}\boldsymbol{\Omega})\right). \tag{12}$$

If all RWs in the system are identical (at least their nominal axial moments of inertia $\widetilde{I}_k = \widetilde{I}$), then:

$$\mathbf{G} = \widetilde{I}\left(\begin{array}{ccc}\mathbf{e}_1 & \dots & \mathbf{e}_n\end{array}\right) = \widetilde{I}\mathbf{A}$$

and (12) becomes:

$$M_k^{int} = -\mathbf{e}_k^T\left(\mathbf{A}\mathbf{A}^T\right)^{-1}\left(\mathbf{M}_{ctrl} + \widetilde{I}\boldsymbol{\omega}\times(\mathbf{A}\boldsymbol{\Omega})\right).$$

### 4.3. Equations of Motion Analysis

Considered controller ensures asymptotic stability of simple system Equation (7), hence we can expect that if $\varepsilon$ is sufficiently small, the satellite motion would be in the vicinity of the required one. Therefore, it is possible to linearize the equations of motion. Then, the attitude quaternion is:

$$\mathbf{Q} = \left(\begin{array}{c}q_0 \\ \mathbf{q}\end{array}\right) \approx \left(\begin{array}{c}1 \\ \frac{1}{2}\boldsymbol{\varphi}\end{array}\right),$$

where $\boldsymbol{\varphi} = \left(\begin{array}{ccc}\varphi_1 & \varphi_2 & \varphi_3\end{array}\right)$ correspond to three Euler rotation angles (sequence 1–2–3 at angles $\varphi_1, \varphi_2, \varphi_3$). From Kinematics (3), we also obtain that the satellite angular velocity in linear approximation is $\boldsymbol{\omega} = \dot{\boldsymbol{\varphi}}$.

The solution of System (8) and linearized kinematics is represented in the power series:

$$\boldsymbol{\omega} = \boldsymbol{\omega}^0 + \varepsilon\boldsymbol{\omega}^1 + \dots$$
$$\Omega_k = \Omega_k^0 + \varepsilon\Omega_k^1 + \dots = \mathbf{e}_k\left(\Omega_k^0 + \varepsilon\Omega_k^1 + \dots\right)$$
$$\mathbf{q} \approx \frac{1}{2}\left(\boldsymbol{\varphi}^0 + \varepsilon\boldsymbol{\varphi}^1 + \dots\right).$$

Therefore, RWs' controls become:

$$\begin{aligned}M_k^{int} &\approx \mathbf{e}_k^T\left(\mathbf{A}\mathbf{A}^T\right)^{-1}\left(k_\omega\left(\boldsymbol{\omega}^0 + \varepsilon\boldsymbol{\omega}^1\right) + \frac{k_q}{2}\left(\boldsymbol{\varphi}^0 + \varepsilon\boldsymbol{\varphi}^1\right)\right)\boldsymbol{\varphi}^1 \\ &= \sum_k\left(\mathbf{u}_k\cos\left(\alpha_k^0 + \Omega_k t\right) + \mathbf{v}_k\sin\left(\alpha_k^0 + \Omega_k t\right)\right) \\ &\quad -\mathbf{e}_k^T\left(\mathbf{A}\mathbf{A}^T\right)^{-1}\left(\left(\boldsymbol{\omega}^0 + \varepsilon\boldsymbol{\omega}^1\right)\times\left(\widetilde{\mathbf{J}}\left(\boldsymbol{\omega}^0 + \varepsilon\boldsymbol{\omega}^1\right) + \sum_k\widetilde{\mathbf{I}}_k\left(\Omega_k^0 + \varepsilon\Omega_k^1\right)\right)\right).\end{aligned}$$

Their substitution in the equations of motion and comparison of terms with the same powers of $\varepsilon$ gives the system for the undisturbed motion (zero approximation):

$$\left(\widetilde{\mathbf{J}} - \sum_k\widetilde{I}_k\mathbf{e}_k\mathbf{e}_k^T\right)\dot{\boldsymbol{\omega}}^0 = -k_\omega\boldsymbol{\omega}^0 - \frac{k_q}{2}\boldsymbol{\varphi}^0,$$

$$\dot{\Omega}_k^0 = \mathbf{e}_k^T\left(\mathbf{A}\mathbf{A}^T\right)^{-1}\left(k_\omega\boldsymbol{\omega}^0 + \frac{k_q}{2}\boldsymbol{\varphi}^0 + \boldsymbol{\omega}^0\times\left(\widetilde{\mathbf{J}}\boldsymbol{\omega}^0 + \sum_k\widetilde{\mathbf{I}}_k\Omega_k^0\right)\right) + \mathbf{e}_k^T\widetilde{\mathbf{I}}_k\left(\widetilde{\mathbf{J}} - \sum_k\widetilde{I}_k\mathbf{e}_k\mathbf{e}_k^T\right)^{-1}\left(k_\omega\boldsymbol{\omega}^0 + \frac{k_q}{2}\boldsymbol{\varphi}^0\right),$$

and equations for the first approximation of the satellite angular velocity:

$$\left(\widetilde{\mathbf{J}} - \sum_k \widetilde{I}_k \mathbf{e}_k \mathbf{e}_k^T\right)\dot{\boldsymbol{\omega}}^1 = -k_\omega \boldsymbol{\omega}^1 - \frac{k_q}{2}\boldsymbol{\varphi}^1$$

$$+ \left[ \delta\mathbf{a}\Big|_{\substack{\boldsymbol{\omega}=\boldsymbol{\omega}^0 \\ \boldsymbol{\Omega}_k = \boldsymbol{\Omega}_k^0}} - \sum_k \left( \delta b_k\Big|_{\substack{\boldsymbol{\omega}=\boldsymbol{\omega}^0 \\ \boldsymbol{\Omega}_k = \boldsymbol{\Omega}_k^0}} \mathbf{e}_k + b_k\Big|_{\substack{\boldsymbol{\omega}=\boldsymbol{\omega}^0 \\ \boldsymbol{\Omega}_k = \boldsymbol{\Omega}_k^0}} \left(\mathbf{E}_{3\times3} - \mathbf{e}_k\mathbf{e}_k^T\right)\frac{\delta\mathbf{I}_k\mathbf{e}_k}{\widetilde{I}_k} \right) - \right.$$

$$\left. - \sum_k \left(\delta\mathbf{I}_k - \delta\mathbf{I}_k\mathbf{e}_k\mathbf{e}_k^T - \mathbf{e}_k\mathbf{e}_k^T\delta\mathbf{I}_k + \mathbf{e}_k\mathbf{e}_k^T\delta\mathbf{I}_k\right)\left(\widetilde{\mathbf{J}} - \sum_k \widetilde{I}_k\mathbf{e}_k\mathbf{e}_k^T\right)^{-1}\left(-k_\omega\boldsymbol{\omega}^0 - k_q\mathbf{q}^0\right) \right]$$

Zero approximation has an asymptotically stable solution $\boldsymbol{\omega}^0 = 0$, $\boldsymbol{\varphi}^0 = 0$. This leads to:

$$\delta b_k\Big|_{\substack{\boldsymbol{\omega}=\boldsymbol{\omega}^0 \\ \boldsymbol{\Omega}_k = \boldsymbol{\Omega}_k^0}} = 0, \quad b_k\Big|_{\substack{\boldsymbol{\omega}=\boldsymbol{\omega}^0 \\ \boldsymbol{\Omega}_k = \boldsymbol{\Omega}_k^0}} = 0, \quad \delta\mathbf{a} = -\sum_k \boldsymbol{\Omega}_k^0 \times \delta\mathbf{I}_k\boldsymbol{\Omega}_k^0$$

In addition, from the second equation of zero approximation, we can see that $\Omega_k^0 \to const$. Therefore, the equations for the first approximation are simplified:

$$\left(\widetilde{\mathbf{J}} - \sum_k \widetilde{I}_k\mathbf{e}_k\mathbf{e}_k^T\right)\dot{\boldsymbol{\omega}}^1 = -k_\omega\boldsymbol{\omega}^1 - \frac{k_q}{2}\boldsymbol{\varphi}^1 - \sum_k \boldsymbol{\Omega}_k^0 \times \delta\mathbf{I}_k\boldsymbol{\Omega}_k^0 \qquad (13)$$

Note that $\delta\mathbf{I}_k$ is not constant in the Body Frame since the imbalanced RWs rotate. It can be represented in the Body Frame as follows:

$$\delta\mathbf{I}_k^{BF} = \mathbf{D}_k^T\delta\mathbf{I}_k^{RW}\mathbf{D}_k$$

where:

$$\mathbf{D}_k = \begin{pmatrix} \cos\alpha_k & \sin\alpha_k & 0 \\ -\sin\alpha_k & \cos\alpha_k & 0 \\ 0 & 0 & 1 \end{pmatrix}\mathbf{D}_k^0, \quad \alpha_k = \int_{t_0}^t \Omega_k dt + \alpha_k(t_0) \qquad (14)$$

is the rotation matrix that corresponds to the current RW position, $\mathbf{D}_k^0$ describes the rotation from the Body Frame to the $k$-th RW Frame in the initial moment, $\delta\mathbf{I}_k^{RW} = const$ is the RW imbalance in $k$-th RW Frame. Since $\Omega_k^0$ is constant in zero approximation, $\mathbf{D}_k$ describes constant rotation and Equation (13) becomes the nonhomogeneous linear system of differential equations:

$$\left(\widetilde{\mathbf{J}} - \sum_k \widetilde{I}_k\mathbf{e}_k\mathbf{e}_k^T\right)\ddot{\boldsymbol{\varphi}}^1 = -k_\omega\dot{\boldsymbol{\varphi}}^1 - \frac{k_q}{2}\boldsymbol{\varphi}^1 + \sum_k \left(\mathbf{g}_k\cos\left(t\Omega_k^0 + \alpha_k^0\right) + \mathbf{f}_k\sin\left(t\Omega_k^0 + \alpha_k^0\right)\right) \qquad (15)$$

which can be solved analytically. Note that $\alpha_k^0$ is a constant initial phase. The value of the phase depends on the actual transient motion before the satellites settle near the required state. The solution of the homogeneous equation converges to zero asymptotically while the partial solution is:

$$\boldsymbol{\varphi}^1 = \sum_k \left(\mathbf{u}_k\cos\left(\alpha_k^0 + \Omega_k t\right) + \mathbf{v}_k\sin\left(\alpha_k^0 + \Omega_k t\right)\right)$$

where:

$$\mathbf{u}_k = -\left(\mathbf{M}^2 + \left(k_\omega\Omega_k^0\right)^2\mathbf{E}_{3\times3}\right)^{-1}\left(k_\omega\Omega_k^0\mathbf{f}_k + \mathbf{M}\mathbf{g}_k\right),$$

$$\mathbf{v}_k = \left(\mathbf{M}^2 + \left(k_\omega\Omega_k^0\right)^2\mathbf{E}_{3\times3}\right)^{-1}\left(k_\omega\Omega_k^0\mathbf{g}_k - \mathbf{M}\mathbf{f}_k\right),$$

$$\mathbf{M} = \left(\left(\widetilde{\mathbf{J}} - \sum_k \widetilde{I}_k\mathbf{e}_k\mathbf{e}_k^T\right)\left(\Omega_k^0\right)^2 - \frac{k_q}{2}\mathbf{E}_{3\times3}\right).$$

The angular velocity for the first approximation is:

$$\boldsymbol{\omega}^1 = \sum_k \Omega_k^0 \left( -\mathbf{u}_k \sin\left( \alpha_k^0 + \Omega_k^0 t \right) + \mathbf{v}_k \cos\left( \alpha_k^0 + \Omega_k^0 t \right) \right) \tag{16}$$

Hence the attitude stabilization error can be estimated by:

$$\Delta\boldsymbol{\omega} \approx \varepsilon \sum_k \Omega_k^0 \left( -\mathbf{u}_k \sin\left( \alpha_k^0 + \Omega_k^0 t \right) + \mathbf{v}_k \cos\left( \alpha_k^0 + \Omega_k^0 t \right) \right) \tag{17}$$

For each *i*-th component, then:

$$\Delta\omega_i \leq \varepsilon \sum_k \Omega_k^0 \sqrt{u_{ki}^2 + v_{ki}^2} \tag{18}$$

This simple estimate might be utilized at the preliminary stage of a spacecraft design to determine the dynamic imbalance requirements for RWs. Each term in Equation (18) is equivalent to:

$$\Delta\omega_{ik} \sim \Omega_k^0 d \tag{19}$$

when $\Omega_k^0$ is large enough, here, $d$ is the dynamic imbalance magnitude.

*4.4. Illustrative Example*

Assume that each RW tensor of inertia in its own Frame is:

$$\mathbf{I}_k^{RW} = \begin{pmatrix} I_{eq} & 0 & 0 \\ 0 & I_{eq} & 0 \\ 0 & 0 & I_{ax} \end{pmatrix} + \Delta\mathbf{I}_k^{RW}, \quad \Delta\mathbf{I}_k^{RW} = \begin{pmatrix} \Delta I_{11}^k & \Delta I_{12}^k & \Delta I_{13}^k \\ \Delta I_{12}^k & \Delta I_{22}^k & \Delta I_{23}^k \\ \Delta I_{13}^k & \Delta I_{23}^k & \Delta I_{33}^k \end{pmatrix},$$

where $I_{eq}$, $I_{ax}$ are the RWs' equatorial and axial moments of inertia (all RW nominal values are supposed to be identical). Note that $\Delta\mathbf{I}_k^{RW}$ is symmetric, and all its components are considered small with respect to the axial moment of inertia.

In RW Frame:

$$\boldsymbol{\Omega}_k^0 \times \mathbf{I}_k \boldsymbol{\Omega}_k^0 = \left( \Omega_k^0 \right)^2 \begin{pmatrix} -\Delta I_{23}^k \\ \Delta I_{13}^k \\ 0 \end{pmatrix}$$

Let for all RWs $\Delta I_{13}^k = 0$, and $\Delta I_{13}^k = -d$ is the dynamic imbalance. Small parameter is introduced as follows:

$$\varepsilon = d / I_{ax}$$

Let the nominal RWs' axes of rotation coincide with $\widetilde{\mathbf{J}}$ principal axes of inertia. Then, in the Body Frame:

$$\boldsymbol{\Omega}_1 = \begin{pmatrix} \Omega_1 & 0 & 0 \end{pmatrix}^T, \quad \boldsymbol{\Omega}_2 = \begin{pmatrix} 0 & \Omega_2 & 0 \end{pmatrix}^T, \quad \boldsymbol{\Omega}_3 = \begin{pmatrix} 0 & 0 & \Omega_3 \end{pmatrix}^T.$$

Matrices that describe the rotation from the RW Frame to the Body Frame are:

$$\mathbf{D}_1 = \begin{pmatrix} 0 & 0 & 1 \\ \sin\alpha_1 & \cos\alpha_1 & 0 \\ -\cos\alpha_1 & \sin\alpha_1 & 0 \end{pmatrix}, \quad \mathbf{D}_2 = \begin{pmatrix} \cos\alpha_2 & -\sin\alpha_2 & 0 \\ 0 & 0 & 1 \\ -\sin\alpha_2 & -\cos\alpha_2 & 0 \end{pmatrix},$$

$$\mathbf{D}_3 = \begin{pmatrix} \cos\alpha_3 & -\sin\alpha_3 & 0 \\ \sin\alpha_3 & \cos\alpha_3 & 0 \\ 0 & 0 & 1 \end{pmatrix}.$$

Here $\alpha_k = \alpha_k^0 + \Omega_k^0 t$, as in the previous section, corresponds to the current RW rotation angle. Finally, the right part of the Equation (15) becomes:

$$\sum_k \boldsymbol{\Omega}_k^0 \times \delta \mathbf{I}_k \boldsymbol{\Omega}_k^0 = \begin{pmatrix} 0 \\ \sin\left(\alpha_1^0 + \Omega_1^0 t\right) \\ -\cos\left(\alpha_1^0 + \Omega_1^0 t\right) \end{pmatrix} \left(\Omega_1^0\right)^2 I_{ax} + \begin{pmatrix} \cos\left(\alpha_2^0 + \Omega_2^0 t\right) \\ 0 \\ -\sin\left(\alpha_2^0 + \Omega_2^0 t\right) \end{pmatrix} \left(\Omega_2^0\right)^2 I_{ax}$$
$$+ \begin{pmatrix} \cos\left(\alpha_3^0 + \Omega_3^0 t\right) \\ \sin\left(\alpha_2^0 + \Omega_2^0 t\right) \\ 0 \end{pmatrix} \left(\Omega_3^0\right)^2 I_{ax}.$$

In this case, vectors $\mathbf{f}_k$ and $\mathbf{g}_k$ in Equation (15) are:

$$\mathbf{f}_1 = \begin{pmatrix} 0 & -1 & 0 \end{pmatrix}^T \left(\Omega_1^0\right)^2 I_{ax}, \quad \mathbf{f}_2 = \begin{pmatrix} 0 & 0 & 1 \end{pmatrix}^T \left(\Omega_2^0\right)^2 I_{ax}, \quad \mathbf{f}_3 = \begin{pmatrix} 0 & -1 & 0 \end{pmatrix}^T \left(\Omega_3^0\right)^2 I_{ax}$$

$$\mathbf{g}_1 = \begin{pmatrix} 0 & 0 & 1 \end{pmatrix}^T \left(\Omega_1^0\right)^2 I_{ax}, \quad \mathbf{g}_2 = \begin{pmatrix} -1 & 0 & 0 \end{pmatrix}^T \left(\Omega_2^0\right)^2 I_{ax}, \quad \mathbf{g}_3 = \begin{pmatrix} -1 & 0 & 0 \end{pmatrix}^T \left(\Omega_3^0\right)^2 I_{ax}$$

For the illustration purposes, consider the evolution of the angular velocity after the transient motion. So, the satellite angular velocity is zero, quaternion is identical and almost all initial angular momentum is stored in the RWs. System parameters are presented in Table 1. The control parameters are $k_\omega = 0.01$ N·m·s, $k_a = 0.001$ N·m. The results are presented in Figures 6–8. The red curves in figures are the numerical solutions, the blue curves are the approximate solutions in Equation (16), the black horizontal lines are the estimations in Equation (18). In order to test a more realistic scenario, we also include gravity gradient torque into the simulation. This torque is rather small, so at sufficiently small time spans, it would not affect the results.

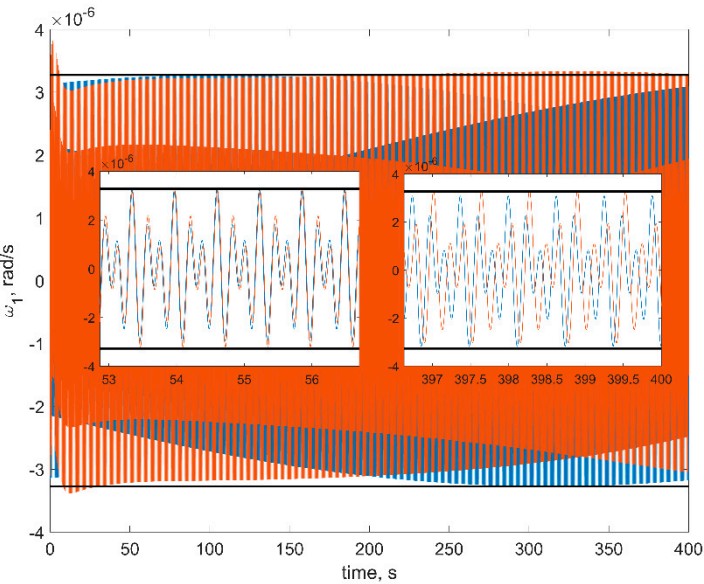

**Figure 6.** Evolution of $\omega_1$ (red is numerical solution, blue is approximate solution, horizontal lines are the estimations).

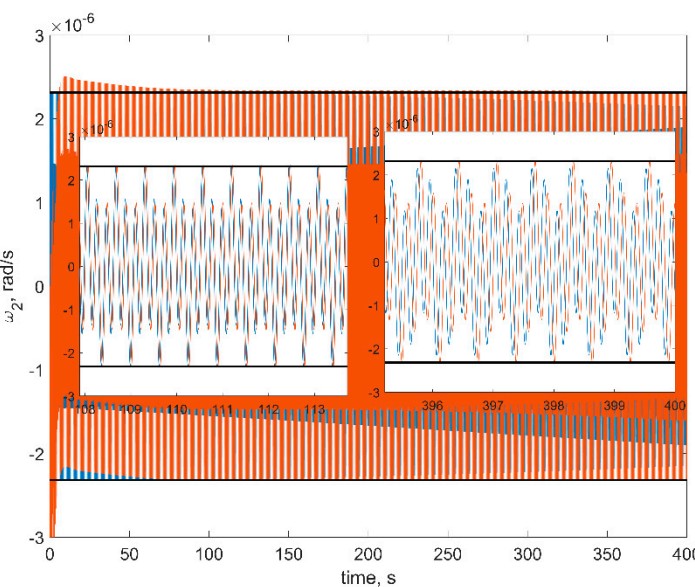

**Figure 7.** Evolution of $\omega_2$ (red is numerical solution, blue is approximate solution, horizontal lines are the estimations).

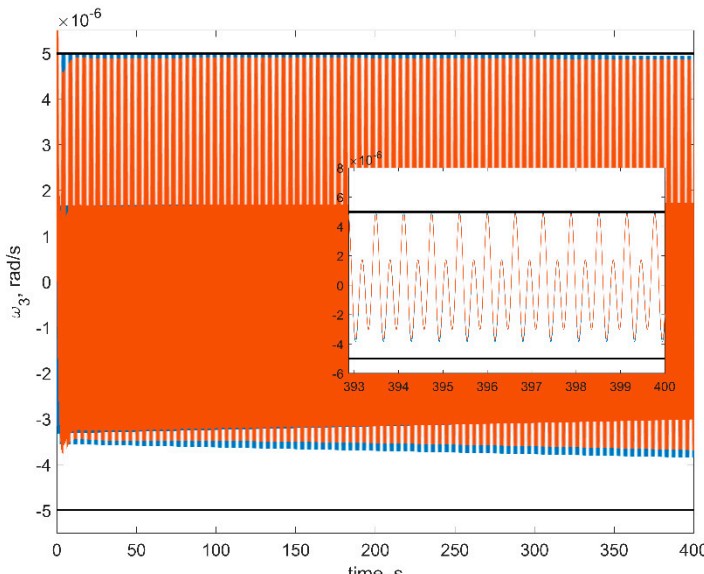

**Figure 8.** Evolution of $\omega_3$ (red is numerical solution, blue is approximate solution, horizontal lines are the estimations).

First of all, one can see from figures that estimations in Equation (18) are in good accordance with the numerical simulation: numerical results are within the estimation borders, and relative difference between the linearized model of motion and the full one is around 3%. These estimations are of the utmost practical interest since these values show the stabilization accuracy of the satellite. The evolution of the angular velocity components is also close to the numerical solution. However, it should be noted that for the first and second components (Figures 6 and 7), the phase difference increases by the end of the time interval. This is due to the non-uniform evolution of the RWs' angles of rotation. Figure 9 illustrates this effect.

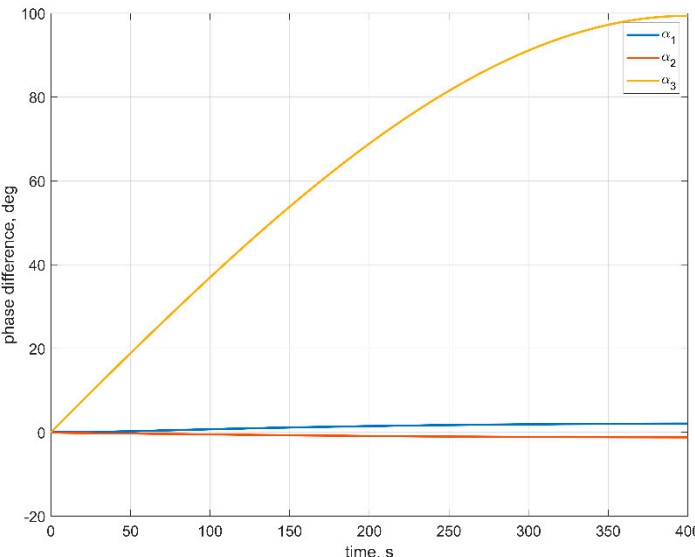

**Figure 9.** Phase difference between uniform and numerical RW angle evolution.

In Figure 9, one can see that the phase difference for the third RW is large, which results in the noticeable difference between the numerical and approximate solutions for the first and second angular velocity components.

The following numerical example is considered for the Relation (19) illustration. The first RW angular velocity is taken as $\Omega_1 = 10$ rad/s for the first case, $\Omega_1 = 50$ rad/s for the second one and $\Omega_2 = \Omega_3 = 0$ for both cases. Other parameters are the same. Results are presented in Figure 10.

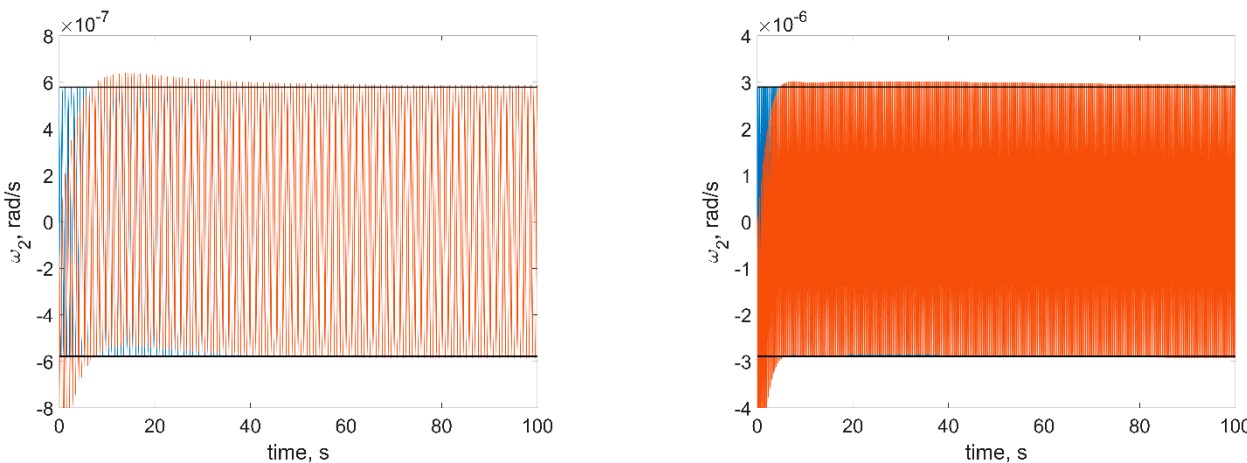

**Figure 10.** Stabilization accuracy for $\Omega_1 = 10$ s $^{-1}$ (**left**) and $\Omega_1 = 50$ s $^{-1}$ (**right**).

From Figure 10, one can see that if $\Omega_1$ increases fivefold then the amplitude of angular velocity oscillations also increases fivefold (the angular velocity amplitude increases by the factor of 5.04). So, the Relation (19) is in good accordance with numerical simulation.

## 5. Effect of Static Imbalance on Inertial Stabilization

We apply the similar technique to the study of the effect of static imbalance. Its value is usually higher than the one for the dynamic imbalance. In addition, it turns out that the stabilization accuracy in this case depends not only on the pure imbalance parameters.

Using the same approach as in Section 3, the following first order approximation equations are obtained (here $\mathbf{I}_k$ are supposed to be axially symmetrical):

$$\left(\widetilde{\mathbf{J}} - \sum_k \widetilde{I}_k \mathbf{e}_k \mathbf{e}_k^T\right)\dot{\boldsymbol{\omega}}^1 = -k_\omega \boldsymbol{\omega}^1 - \frac{k_q}{2}\boldsymbol{\varphi}^1 - \sum_k m_k \boldsymbol{\rho}_k \times \boldsymbol{\Omega}_k^0 \times \boldsymbol{\Omega}_k^0 \times \delta\boldsymbol{\rho}_{kc} \qquad (20)$$

Here $\delta\boldsymbol{\rho}_{kc}$ represents the vector from point $O_k$ to the RW center of mass and depends on RW rotation angle:

$$\delta\boldsymbol{\rho}_{kc}^{BF} = \mathbf{D}_k \delta\boldsymbol{\rho}_{kc}^{RW}$$

where $\mathbf{D}_k$ is determined by Equation (14). The resulting equations are similar to Equation (15), so the resulting equations of motion are also similar.

$$\Delta\boldsymbol{\omega} \approx \rho_k \delta\rho_{kc} \sum_k \Omega_k^0\left(-\mathbf{u}_k \sin\left(\alpha_k^0 + \Omega_k^0 t\right) + \mathbf{v}_k \cos\left(\alpha_k^0 + \Omega_k^0 t\right)\right) \qquad (21)$$

Note that attitude stabilization accuracy depends on the RW position with respect to the system center of mass $\boldsymbol{\rho}_k$, i.e., the farther RWs are placed from the system center of mass, the worse stabilization accuracy is. This result is especially useful as it allows us to reduce the effect of vibrations caused by static imbalance at the early stage of satellite design.

Consider the same illustrative example as in Section 4.4. The following parameters are taken additionally: $\boldsymbol{\rho}_1 = \begin{pmatrix} 0 & 0 & 0.01 \end{pmatrix}^T$ m, $\boldsymbol{\rho}_2 = \begin{pmatrix} 0.01 & 0 & 0 \end{pmatrix}^T$ m, $\boldsymbol{\rho}_2 = \begin{pmatrix} 0.01 & 0 & 0 \end{pmatrix}^T$ m and $\boldsymbol{\rho}_{kc} = \begin{pmatrix} \frac{s}{m_k} & 0 & 0 \end{pmatrix}^T$ ($s$ is the static imbalance, see Table 1) for all RWs. All other parameters are the same (without dynamic imbalance). This leads to:

$$\sum_k m_k \boldsymbol{\rho}_k \times \boldsymbol{\Omega}_k^0 \times \boldsymbol{\Omega}_k^0 \times \boldsymbol{\rho}_{kc} = s\rho_1 \left(\Omega_1^0\right)^2 \begin{pmatrix} \sin\left(\alpha_1^0 + \Omega_1^0 t\right) \\ 0 \\ 0 \end{pmatrix} + s\rho_2 \left(\Omega_2^0\right)^2 \begin{pmatrix} 0 \\ -\sin\left(\alpha_2^0 + \Omega_2^0 t\right) \\ 0 \end{pmatrix} +$$
$$+ s\rho_3 \left(\Omega_3^0\right)^2 \begin{pmatrix} 0 \\ 0 \\ \cos\left(\alpha_2^0 + \Omega_2^0 t\right) \end{pmatrix}.$$

In this case vector $\mathbf{f}_k$ and $\mathbf{g}_k$ in Equation (15) are:

$$\mathbf{f}_1 = \begin{pmatrix} -1 & 0 & 0 \end{pmatrix}^T \left(\Omega_1^0\right)^2 s\rho_1, \quad \mathbf{f}_2 = \begin{pmatrix} 0 & 1 & 0 \end{pmatrix}^T \left(\Omega_2^0\right)^2 s\rho_2, \quad \mathbf{f}_3 = \begin{pmatrix} 0 & 0 & 0 \end{pmatrix}^T$$

$$\mathbf{g}_1 = \begin{pmatrix} 0 & 0 & 0 \end{pmatrix}^T, \quad \mathbf{g}_2 = \begin{pmatrix} 0 & 0 & 0 \end{pmatrix}^T, \quad \mathbf{g}_3 = \begin{pmatrix} 0 & 0 & -1 \end{pmatrix}^T \left(\Omega_3^0\right)^2 s\rho_3$$

The results of illustrative numerical simulation are presented in Figures 11–13. Black lines are the analytical estimations, red and blue lines are the numerical simulation results.

The example shows that the Expressions (21) are in a good accordance with numerical simulations. As one can see from the close-ups, the difference between analytical estimations and numerical simulation results is rather small and is constrained to 5% in the worst case. The difference can be explained by the gravity gradient torque which is included in the numerical model but is not taken into account in the analytical study.

As one can see, both dynamic and static imbalances lead to the additional terms in the right parts of Equations (15) and (20), which do not depend on satellite state vector components for the first order approximation, so the total stabilization accuracy is the sum of Equations (17) and (21).

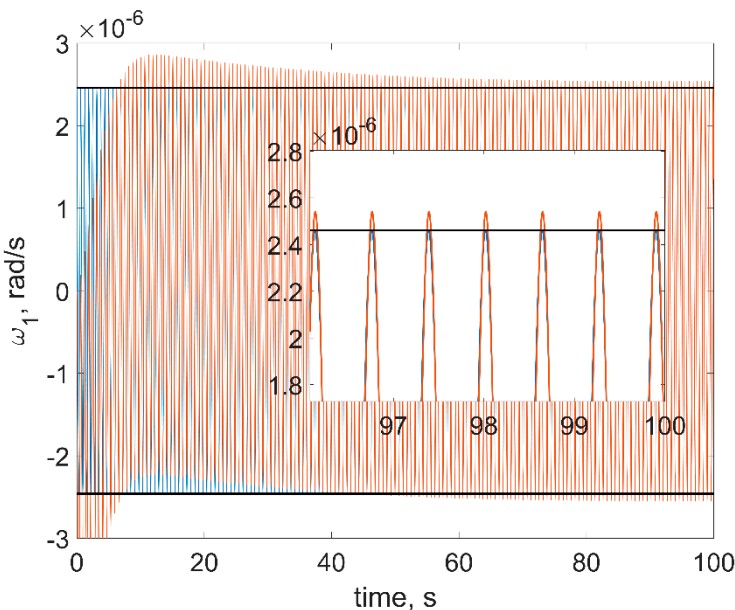

**Figure 11.** Evolution of $\omega_1$ for static imbalance case (red is numerical solution, blue is approximate solution, horizontal lines are the estimations).

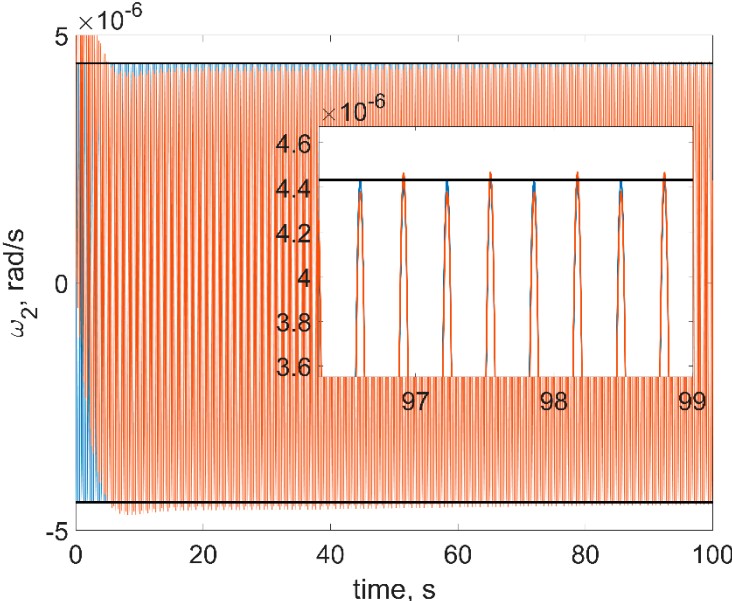

**Figure 12.** Evolution of $\omega_2$ for static imbalance case (red is numerical solution, blue is approximate solution, horizontal lines are the estimations).

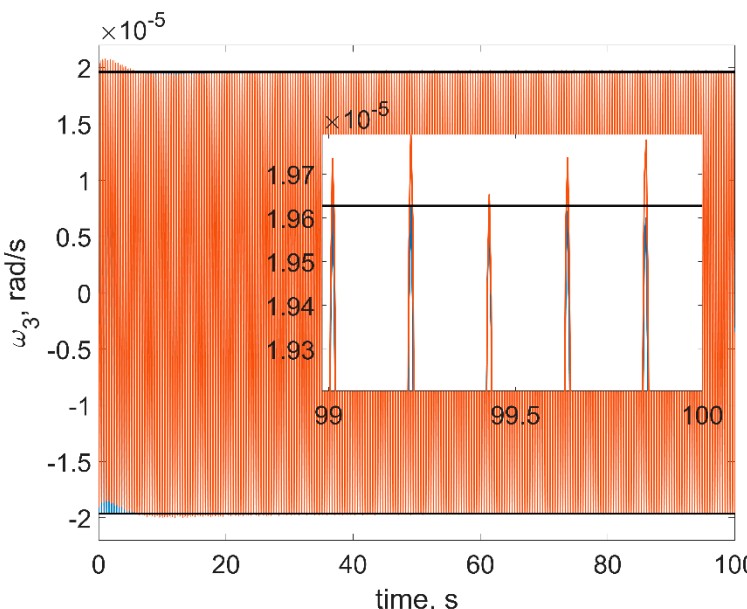

**Figure 13.** Evolution of $\omega_3$ for static imbalance case (red is numerical solution, blue is approximate solution, horizontal lines are the estimations).

## 6. Conclusions

In this paper, the satellite motion analysis is carried out. The model includes RWs' static and dynamic imbalances along with coupling orbital and angular satellite motion with RW rotation. Software implementation of the model is validated using the momentum, angular momentum, and kinetic energy conservation laws. The simulations show that the integration step should be rather small due to the high values of typical RW angular velocities. This fact makes purely numerical analysis difficult. In order to solve this problem, the analytical approximations for the satellite stabilization accuracy are obtained in closed form for the static and dynamic imbalances' presence in the inertial stabilization case. The comparison of the numerical simulation and approximate solution shows that they are in a good accordance (relative error is about several percent). The explicit expressions can easily be implemented and are useful during the preliminary satellite design stage.

The estimations of the attitude accuracy are obtained for the case of the satellite's inertial stabilization. The case of orbital stabilization is the main goal of future research.

**Author Contributions:** Results analysis, D.I.; original draft preparation, D.R.; model of motion verification, Y.M.; software implementation, S.T.; work coordination, M.O. All authors have read and agreed to the published version of the manuscript.

**Funding:** This research received no external funding.

**Institutional Review Board Statement:** Not applicable.

**Informed Consent Statement:** Not applicable.

**Data Availability Statement:** Not applicable.

**Conflicts of Interest:** The authors declare no conflict of interest.

**Appendix A**

In order to derive equations of motion, the general equation of dynamics is used. Terms for each independent virtual displacements are:

$$\delta \mathbf{R}_O : \sum_i \left( m_i \ddot{\mathbf{R}}_i - \mathbf{F}_i \right) + \sum_k \sum_j \left( m_{kj} \ddot{\mathbf{R}}_{kj} - \mathbf{F}_{kj} \right) = 0,$$

$$\delta \boldsymbol{\theta} : \sum_i \mathbf{r}_i \times \left( m_i \ddot{\mathbf{R}}_i - \mathbf{F}_i \right) + \sum_k \sum_j \left( \boldsymbol{\rho}_k + \boldsymbol{\rho}_{kj} \right) \times \left( m_{kj} \ddot{\mathbf{R}}_{kj} - \mathbf{F}_{kj} \right) = 0, \tag{A1}$$

$$\delta \varphi_k : \mathbf{e}_k^T \left( \sum_j \boldsymbol{\rho}_{kj} \times \left( m_{kj} \ddot{\mathbf{R}}_{kj} - \mathbf{F}_{kj} \right) \right) - M_k^{int} = 0.$$

We start with the equations of motion for point $O$ (first equation). Since:

$$\ddot{\mathbf{R}}_i = \ddot{\mathbf{R}}_O + \dot{\boldsymbol{\omega}} \times \mathbf{r}_i + \boldsymbol{\omega} \times \boldsymbol{\omega} \times \mathbf{r}_i,$$

$$\ddot{\mathbf{R}}_{kj} = \ddot{\mathbf{R}}_O + \dot{\boldsymbol{\omega}} \times \boldsymbol{\rho}_k + \boldsymbol{\omega} \times \boldsymbol{\omega} \times \boldsymbol{\rho}_k + \left( \dot{\boldsymbol{\omega}} + \dot{\boldsymbol{\Omega}}_k + \boldsymbol{\omega} \times \boldsymbol{\Omega}_k \right) \times \boldsymbol{\rho}_{kj} + (\boldsymbol{\omega} + \boldsymbol{\Omega}_k) \times (\boldsymbol{\omega} + \boldsymbol{\Omega}_k) \times \boldsymbol{\rho}_{kj},$$

it can be rewritten as follows:

$$\left( m_s + \sum_k m_k \right) \ddot{\mathbf{R}}_O + \dot{\boldsymbol{\omega}} \times \left( m_s \mathbf{r}_s + \sum_k m_k (\boldsymbol{\rho}_k + \boldsymbol{\rho}_{kc}) \right) + \sum_k m_k \left( \dot{\boldsymbol{\Omega}}_k + \boldsymbol{\omega} \times \boldsymbol{\Omega}_k \right) \times \boldsymbol{\rho}_{kc} +$$

$$+ \boldsymbol{\omega} \times \boldsymbol{\omega} \times \left( m_s \mathbf{r}_s + \sum_k m_k \boldsymbol{\rho}_k \right) + \sum_k m_k (\boldsymbol{\omega} + \boldsymbol{\Omega}_k) \times (\boldsymbol{\omega} + \boldsymbol{\Omega}_k) \times \boldsymbol{\rho}_{kc} = \mathbf{F}_s + \sum_k \mathbf{F}_k.$$

where:

$$\mathbf{r}_s = \frac{\sum_i \mathbf{r}_i m_i}{m_s}, \quad \boldsymbol{\rho}_{kc} = \frac{\sum_j \boldsymbol{\rho}_{kj} m_j}{m_k}, \quad m_k = \sum_j m_{kj}, \quad m_s = \sum_i m_i, \quad \mathbf{F}_s = \sum_i \mathbf{F}_i, \quad \mathbf{F}_k = \sum_j \mathbf{F}_{kj}.$$

Since point $O$ is an arbitrary fixed hull point, it is reasonable to choose it, so:

$$m_s \mathbf{r}_s + \sum_k m_k \boldsymbol{\rho}_k = 0 \tag{A2}$$

Then:

$$m \ddot{\mathbf{R}}_O + \sum_k m_k \left( \dot{\boldsymbol{\omega}} + \dot{\boldsymbol{\Omega}}_k + \boldsymbol{\omega} \times \boldsymbol{\Omega}_k \right) \times \boldsymbol{\rho}_{kc} + \sum_k m_k (\boldsymbol{\omega} + \boldsymbol{\Omega}_k) \times (\boldsymbol{\omega} + \boldsymbol{\Omega}_k) \times \boldsymbol{\rho}_{kc} = \mathbf{F}_s + \sum_k \mathbf{F}_k.$$

Introducing:

$$\mathbf{M}_s = \sum_i \mathbf{r}_i \times \mathbf{F}_i, \quad \mathbf{M}_k = \sum_j \boldsymbol{\rho}_{kj} \times \mathbf{F}_{kj},$$

$$\mathbf{J}_s = -\sum_i [\mathbf{r}_i]_\times [\mathbf{r}_i]_\times m_i, \quad \mathbf{I}_k = -\sum_j \left[ \boldsymbol{\rho}_{kj} \right]_\times \left[ \boldsymbol{\rho}_{kj} \right]_\times m_{kj}.$$

and utilizing properties:

$$\mathbf{a} \times (\boldsymbol{\omega} \times \mathbf{b}) = -\mathbf{a} \times (\mathbf{b} \times \boldsymbol{\omega}) = -[\mathbf{a}]_\times [\mathbf{b}]_\times \boldsymbol{\omega},$$

$$\mathbf{a} \times (\boldsymbol{\omega} \times (\boldsymbol{\omega} \times \mathbf{a})) = -\boldsymbol{\omega} \times (\mathbf{a} \times (\mathbf{a} \times \boldsymbol{\omega})) = -\boldsymbol{\omega} \times \left( [\mathbf{a}]_\times [\mathbf{a}]_\times \boldsymbol{\omega} \right),$$

$$\mathbf{a} \times (\boldsymbol{\omega} \times (\boldsymbol{\omega} \times \mathbf{b})) + \mathbf{b} \times (\boldsymbol{\omega} \times (\boldsymbol{\omega} \times \mathbf{a})) = -\boldsymbol{\omega} \times \left( \left( [\mathbf{a}]_\times [\mathbf{b}]_\times + [\mathbf{b}]_\times [\mathbf{a}]_\times \right) \boldsymbol{\omega} \right),$$

equations of hull angular motion are derived from the second Equation of (A1):

$$
\left(m_s + \sum_k (\boldsymbol{\rho}_k + \boldsymbol{\rho}_{kc})m_k\right) \times \ddot{\mathbf{R}}_O +
$$
$$
+ \left(\mathbf{J}_s + \sum_k (-m_k[\boldsymbol{\rho}_k]_\times[\boldsymbol{\rho}_{kc}]_\times - m_k[\boldsymbol{\rho}_{kc}]_\times[\boldsymbol{\rho}_k]_\times - m_k[\boldsymbol{\rho}_k]_\times[\boldsymbol{\rho}_k]_\times + \mathbf{I}_k)\right)\dot{\boldsymbol{\omega}} +
$$
$$
+ \sum_k (-m_k[\boldsymbol{\rho}_k]_\times[\boldsymbol{\rho}_{kc}]_\times + \mathbf{I}_k)\dot{\boldsymbol{\Omega}}_k + \boldsymbol{\omega} \times \left(\mathbf{J}_s - \sum_k m_k[\boldsymbol{\rho}_k]_\times[\boldsymbol{\rho}_k]_\times\right)\boldsymbol{\omega} +
$$
$$
+ \sum_k (m_k\boldsymbol{\rho}_{kc} \times \boldsymbol{\omega} \times \boldsymbol{\omega} \times \boldsymbol{\rho}_k + (-m_k[\boldsymbol{\rho}_k]_\times[\boldsymbol{\rho}_{kc}]_\times + \mathbf{I}_k)(\boldsymbol{\omega} \times \boldsymbol{\Omega}_k) +
$$
$$
+ m_k\boldsymbol{\rho}_k \times (\boldsymbol{\omega} + \boldsymbol{\Omega}_k) \times (\boldsymbol{\omega} + \boldsymbol{\Omega}_k) \times \boldsymbol{\rho}_{kc} + (\boldsymbol{\omega} + \boldsymbol{\Omega}_k) \times \mathbf{I}_k(\boldsymbol{\omega} + \boldsymbol{\Omega}_k))
$$
$$
= \mathbf{M}_s + \sum_k (\mathbf{M}_k + \boldsymbol{\rho}_k \times \mathbf{F}_k).
$$

Under Constraint (A2) and using total satellite tensor of inertia:

$$
\mathbf{J} = \mathbf{J}_s + \sum_k (-m_k[\boldsymbol{\rho}_k]_\times[\boldsymbol{\rho}_{kc}]_\times - m_k[\boldsymbol{\rho}_{kc}]_\times[\boldsymbol{\rho}_k]_\times - m_k[\boldsymbol{\rho}_k]_\times[\boldsymbol{\rho}_k]_\times + \mathbf{I}_k)
$$

it becomes:

$$
\sum_k m_k\boldsymbol{\rho}_{kc} \times \ddot{\mathbf{R}}_O + \mathbf{J}\dot{\boldsymbol{\omega}} + \boldsymbol{\omega} \times \mathbf{J}\boldsymbol{\omega} + \sum_k (-m_k[\boldsymbol{\rho}_k]_\times[\boldsymbol{\rho}_{kc}]_\times + \mathbf{I}_k)\dot{\boldsymbol{\Omega}}_k +
$$
$$
+ \sum_k ((-m_k[\boldsymbol{\rho}_k]_\times[\boldsymbol{\rho}_{kc}]_\times + \mathbf{I}_k)(\boldsymbol{\omega} \times \boldsymbol{\Omega}_k) + m_k\boldsymbol{\rho}_k \times \boldsymbol{\Omega}_k \times \boldsymbol{\omega} \times \boldsymbol{\rho}_{kc} + m_k\boldsymbol{\rho}_k \times \boldsymbol{\omega} \times \boldsymbol{\Omega}_k \times \boldsymbol{\rho}_{kc} +
$$
$$
+ m_k\boldsymbol{\rho}_k \times \boldsymbol{\Omega}_k \times \boldsymbol{\Omega}_k \times \boldsymbol{\rho}_{kc} + \boldsymbol{\Omega}_k \times \mathbf{I}_k\boldsymbol{\omega} + \boldsymbol{\omega} \times \mathbf{I}_k\boldsymbol{\Omega}_k + \boldsymbol{\Omega}_k \times \mathbf{I}_k\boldsymbol{\Omega}_k) = \mathbf{M}_s + \sum_k (\mathbf{M}_k + \boldsymbol{\rho}_k \times \mathbf{F}_k).
$$

From the third equation of (A1), we derive:

$$
\mathbf{e}_k^T \left(m_k\boldsymbol{\rho}_{kc} \times \left(\ddot{\mathbf{R}}_O + \dot{\boldsymbol{\omega}} \times \boldsymbol{\rho}_k + \boldsymbol{\omega} \times \boldsymbol{\omega} \times \boldsymbol{\rho}_k\right)\right)
$$
$$
+ \mathbf{e}_k^T \sum_j m_{kj}\boldsymbol{\rho}_{kj} \times \left(\left(\dot{\boldsymbol{\omega}} + \dot{\boldsymbol{\Omega}}_k + \boldsymbol{\omega} \times \boldsymbol{\Omega}_k\right) \times \boldsymbol{\rho}_{kj} + (\boldsymbol{\omega} + \boldsymbol{\Omega}_k) \times (\boldsymbol{\omega} + \boldsymbol{\Omega}_k) \times \boldsymbol{\rho}_{kj}\right)
$$
$$
= M_k^{int} + \mathbf{e}_k^T \mathbf{M}_k
$$

After simplification:

$$
\mathbf{e}_k^T \left(m_k\boldsymbol{\rho}_{kc} \times \left(\ddot{\mathbf{R}}_O + \dot{\boldsymbol{\omega}} \times \boldsymbol{\rho}_k + \boldsymbol{\omega} \times \boldsymbol{\omega} \times \boldsymbol{\rho}_k\right) + \mathbf{I}_k\left(\dot{\boldsymbol{\omega}} + \dot{\boldsymbol{\Omega}}_k + \boldsymbol{\omega} \times \boldsymbol{\Omega}_k\right) + (\boldsymbol{\omega} + \boldsymbol{\Omega}_k) \times \mathbf{I}_k(\boldsymbol{\omega} + \boldsymbol{\Omega}_k)\right)
$$
$$
= M_k^{int} + \mathbf{e}_k^T \mathbf{M}_k
$$

**Appendix B**

We consider the system of equations:

$$
\begin{cases}
\left(\widetilde{\mathbf{J}} + \varepsilon \sum_k \delta\mathbf{I}_k\right)\dot{\boldsymbol{\omega}} + \sum_k \left(\widetilde{\mathbf{I}}_k + \varepsilon\delta\mathbf{I}_k\right)\dot{\boldsymbol{\Omega}}_k = \mathbf{a} + \varepsilon\delta\mathbf{a}, \\
\mathbf{e}_k^T \left(\widetilde{\mathbf{I}}_k + \varepsilon\delta\mathbf{I}_k\right)\left(\dot{\boldsymbol{\omega}} + \dot{\boldsymbol{\Omega}}_k\right) = b_k + \varepsilon\delta b_k, \\
\widetilde{\mathbf{I}}_k\mathbf{e}_k = \widetilde{I}_k\mathbf{e}_k, \quad \mathbf{e}_k^T\widetilde{\mathbf{I}}_k\mathbf{e}_k = \widetilde{I}_k, \quad \mathbf{e}_k^T\delta\mathbf{I}_k\mathbf{e}_k = \delta I_k,
\end{cases} \tag{A3}
$$

and want to eliminate small parameter $\varepsilon$ near the highest order derivatives $\dot{\boldsymbol{\omega}}$, $\dot{\boldsymbol{\Omega}}_k$. Since $\boldsymbol{\Omega}_k = \mathbf{e}_k\Omega_k$, from the second Equation:

$$
\begin{aligned}
\dot{\Omega}_k &= \frac{b_k + \varepsilon\delta b_k - \mathbf{e}_k^T\left(\widetilde{\mathbf{I}}_k + \varepsilon\delta\mathbf{I}_k\right)\dot{\boldsymbol{\omega}}}{\left(\widetilde{I}_k + \varepsilon\delta I_k\right)} \\
&= \frac{1}{\widetilde{I}_k}\left(1 - \varepsilon\frac{\delta I_k}{\widetilde{I}_k}\right)\left(b_k + \varepsilon\delta b_k - \mathbf{e}_k^T\left(\widetilde{\mathbf{I}}_k + \varepsilon\delta\mathbf{I}_k\right)\dot{\boldsymbol{\omega}}\right) + O(\varepsilon^2).
\end{aligned} \tag{A4}
$$

Substitution in the first equation of (A3) yields:

$$
\left( \widetilde{\mathbf{J}} + \varepsilon \sum_k \delta \mathbf{I}_k - \sum_k \left( \frac{1}{\widetilde{I}_k} \left( 1 - \varepsilon \frac{\delta I_k}{\widetilde{I}_k} \right) \left( \widetilde{\mathbf{I}}_k + \varepsilon \delta \mathbf{I}_k \right) \mathbf{e}_k \mathbf{e}_k^T \left( \widetilde{\mathbf{I}}_k + \varepsilon \delta \mathbf{I}_k \right) \right) \right) \dot{\boldsymbol{\omega}}
$$

$$
+ \sum_k \left( \widetilde{\mathbf{I}}_k + \varepsilon \delta \mathbf{I}_k \right) \mathbf{e}_k \frac{1}{\widetilde{I}_k} \left( 1 - \varepsilon \frac{\delta I_k}{\widetilde{I}_k} \right) (b_k + \varepsilon \delta b_k) = \mathbf{a} + \varepsilon \delta \mathbf{a}.
\tag{A5}
$$

Second row can be rewritten as follows:

$$
\mathbf{a} + \varepsilon \delta \mathbf{a} - \sum_k \left( \widetilde{\mathbf{I}}_k + \varepsilon \delta \mathbf{I}_k \right) \mathbf{e}_k \frac{1}{\widetilde{I}_k} \left( 1 - \varepsilon \frac{\delta I_k}{\widetilde{I}_k} \right) (b_k + \varepsilon \delta b_k)
$$

$$
= \mathbf{a} - \sum_k \mathbf{e}_k b_k + \varepsilon \left[ \sum_k \left( -\frac{\delta \mathbf{I}_k \mathbf{e}_k}{\widetilde{I}_k} b_k + \frac{\delta I_k}{\widetilde{I}_k} b_k \mathbf{e}_k - \mathbf{e}_k \delta b_k \right) + \delta \mathbf{a} \right] + O(\varepsilon^2)
$$

Note that:

$$
\widetilde{\mathbf{J}} + \varepsilon \sum_k \delta \mathbf{I}_k - \sum_k \frac{1}{\widetilde{I}_k} \left( 1 - \varepsilon \frac{\delta I_k}{\widetilde{I}_k} \right) \left( \widetilde{\mathbf{I}}_k + \varepsilon \delta \mathbf{I}_k \right) \mathbf{e}_k \mathbf{e}_k^T \left( \widetilde{\mathbf{I}}_k + \varepsilon \delta \mathbf{I}_k \right)
$$

$$
= \widetilde{\mathbf{J}} - \sum_k \widetilde{I}_k \mathbf{e}_k \mathbf{e}_k^T + \varepsilon \sum_k \left( \delta \mathbf{I}_k + \mathbf{e}_k \mathbf{e}_k^T \delta I_k - \mathbf{e}_k \mathbf{e}_k^T \delta I_k - \delta \mathbf{I}_k \mathbf{e}_k \mathbf{e}_k^T \right)
$$

$$
= \left( \mathbf{E}_{3\times3} + \left[ \varepsilon \sum_k \left( \delta \mathbf{I}_k + \mathbf{e}_k \mathbf{e}_k^T \delta I_k - \mathbf{e}_k \mathbf{e}_k^T \delta I_k - \delta \mathbf{I}_k \mathbf{e}_k \mathbf{e}_k^T \right) \right] \left( \widetilde{\mathbf{J}} - \sum_k \widetilde{I}_k \mathbf{e}_k \mathbf{e}_k^T \right)^{-1} \right) \left( \widetilde{\mathbf{J}} - \sum_k \widetilde{I}_k \mathbf{e}_k \mathbf{e}_k^T \right)
$$

Using:

$$
\left( \mathbf{E}_{3\times3} + \varepsilon \mathbf{A} \right)^{-1} = \mathbf{E}_{3\times3} - \varepsilon \mathbf{A} + O\left( \varepsilon^2 \right), \ \varepsilon \ll 1
$$

Equation (A5) becomes:

$$
\left( \widetilde{\mathbf{J}} - \sum_k \widetilde{I}_k \mathbf{e}_k \mathbf{e}_k^T \right) \dot{\boldsymbol{\omega}}
$$

$$
= \left( \mathbf{a} - \sum_k \mathbf{e}_k b_k \right)
$$

$$
- \varepsilon \sum_k \left( \delta \mathbf{I}_k + \mathbf{e}_k \mathbf{e}_k^T \delta I_k - \mathbf{e}_k \mathbf{e}_k^T \delta I_k - \delta \mathbf{I}_k \mathbf{e}_k \mathbf{e}_k^T \right) \left( \widetilde{\mathbf{J}} - \sum_k \widetilde{I}_k \mathbf{e}_k \mathbf{e}_k^T \right)^{-1} \left( \mathbf{a} - \sum_k \mathbf{e}_k b_k \right)
$$

$$
+ \varepsilon \left[ \delta \mathbf{a} - \sum_k \left( \frac{\delta \mathbf{I}_k \mathbf{e}_k}{\widetilde{I}_k} b_k - \frac{\delta I_k}{\widetilde{I}_k} b_k \mathbf{e}_k + \mathbf{e}_k \delta b_k \right) \right]
$$

$$
= \mathbf{c} + \varepsilon \delta \mathbf{c}
$$

Coming back to (A4) yields:

$$
\widetilde{I}_k \dot{\Omega}_k = b_k - \widetilde{I}_k \mathbf{e}_k^T \left( \widetilde{\mathbf{J}} - \sum_k \widetilde{I}_k \mathbf{e}_k \mathbf{e}_k^T \right)^{-1} \mathbf{c}
$$

$$
+ \varepsilon \left( \delta b_k - \frac{\delta I_k}{\widetilde{I}_k} b_k \right)
$$

$$
+ \varepsilon \delta I_k \mathbf{e}_k^T \left( \widetilde{\mathbf{J}} - \sum_k \widetilde{I}_k \mathbf{e}_k \mathbf{e}_k^T \right)^{-1} \mathbf{c} \ - \varepsilon \mathbf{e}_k^T \delta \mathbf{I}_k \left( \widetilde{\mathbf{J}} - \sum_k \widetilde{I}_k \mathbf{e}_k \mathbf{e}_k^T \right)^{-1} \mathbf{c}
$$

$$
- \varepsilon \widetilde{I}_k \mathbf{e}_k^T \left( \widetilde{\mathbf{J}} - \sum_k \widetilde{I}_k \mathbf{e}_k \mathbf{e}_k^T \right)^{-1} \delta \mathbf{c}.
$$

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
