# Peer review of "Effect of Reaction Wheel Imbalances on Attitude and Stabilization Accuracy"

_aerospace, doi:10.3390/aerospace8090252_

Round 1

Reviewer 1 Report

This paper deals with the stabilization error due to reaction wheel imbalances. The derivation is based on a general formulation of the attitude motion of satellite with reaction wheels. The proposed approximation would be useful for system requirements of attitude system and deriving more precise attitude control law. However, this paper have some unclear explanations and the context of research purpose is not imapctive. Thus this paper needs revision to be published in the journal.

I have the following comments/questions:

Major: 

  • In Section 3.2, the explanation of simulation condition is not enough. The formulation describes both translational motion and rotational motion, but Table 2 summarizes the rotational condition only. What is the condition of translational motion? Does the satellite move in an orbit? If so, orbital elements should be clearly described.
  • In Section 3.2, the authors investigate the momentum conservation. For example, Fig. 3 shows the value is less than 3x10^-4. Is this quantitatively reasonable? There is no criteria for the simulation accuracy the authors want to achieve.
  • Also in the simulation results in Section 3.2, the authors claim that the computational burden is high. However modern computers have enough computational power, and the computation for 400s time span is not problematic. The authors claim that the proposed approximation is useful for satellite design stage. In the design stage, the computation is implemented not onboard of satellite, but on the computer on the ground. Thus the motivation of the research purpose seems not impactive. Please add explanations for the advantages of the proposed approximation method.
  • The reaction wheel imbalance seems to have the same magnitude. Is the proposed approximation applicable even if each reaction wheel has different magnitude of imbalances?
  • In Section 4.4, the simulation result after the transient motion is shown. How does the angular velocity behave if the numerical simulation is implemented including transient motion (i.e., the initial condition is non-zero angular rate)?
  • Line 423 says the gravity gradient is included in the numerical simulation. However this is a problem. There is no description about the gravity gradient and the gravity gradient is not included in the formulation of the analytical approximation. Including the gravity gradient makes the result unclear, and the gravity gradient should not be included. If the gravity gradient needs to be included, the analytical approximation also needs to include the effect of the gravity gradient.

Minor:

  • In Fig.2 the mass point should be illustrated at the end of the vector r_i for clearness. Also the three axes of the coordinate frame is necessary at the origin of the vector R_o.
  • At line 114, the definition of \rho_k = OO_k is unclear, and its definition should be clearly explained.
  • Also at line 114, the definition of index i is explained. However, the index i already appears in Eq. (1), and thus the definition should be stated after Eq.(1).
  • At line 116, the author states the degrees of freedom, but it is unclear and not related to the formulation directly.
  • At line 168, the (2,1) component of the matrix S needs minus (because of symmetric matrix).
  • In Table 2 and the other figures, the unit of the angular velocity should be rad/s, not sec^-1.

Round 2

Reviewer 1 Report

The manuscript is revised properly.